# Information Flow Reveals When to Trust Language Models

Rui Xu [1]   Yi Chen [2]   Jiujiu Chen [1]   Sihong Xie [1]

## Abstract

In retrieval-augmented generation, language models can generate incorrect responses if they fail to utilize query-relevant content from the retrieved evidence. This shifts the focus of uncertainty quantification (UQ) toward assessing contextual grounding, i.e., whether predictions are supported by query-relevant tokens. Recent UQ methods unpack language models to characterize how inputs are processed. Nevertheless, these methods focus on a few layers and overlook the whole progressive propagation within the model, thereby failing to fully capture the grounding dynamics essential for reliable uncertainty estimation. We use information flow to build a layer-wise trace that reveals each context token's contribution to the output, providing an interpretable basis for assessing reliability. From this analysis, we introduce two measures to calibrate prediction confidence. The first, *simulatability*, posits that a prediction is more likely to be correct when context token contributions align closely with their true relevance. The second, *concentration*, asserts that a response is more likely to be correct when it is derived from a narrow, focused subset of tokens. Experiments show that our method achieves an average AUROC of 0.709, exceeding the runner-up performance of 0.676, while maintaining moderate computational cost.

## 1. Introduction

Large language models (LLMs), which are predominantly based on the Transformer architecture (Vaswani et al., 2017), have emerged as a transformative technology for a wide range of applications, including question answering (QA) (Tan et al., 2023), summarization (Zhang et al., 2024), and classification (Sun et al., 2023). To equip LLMs with up-to-date and domain-specific knowledge, retrieval-augmented generation (RAG) retrieves relevant documents for a query and incorporates them as additional context to guide generation, as illustrated in Figure 1 (a) (Rodriguez & Boyd-Graber, 2021; Fan et al., 2024; Zhao et al., 2026).

Nevertheless, LLMs cannot consistently generate reliable responses when they struggle to identify and utilize the information that is truly relevant to a given query, even when such information is present in the context (Liu et al., 2024; Iratni et al., 2025). Consequently, effective uncertainty quantification (UQ) should not only diagnose prediction confidence (Fadeeva et al., 2024) and semantic variability (Qiu & Miikkulainen, 2024), but also assess whether the model's prediction is supported by query-relevant context tokens, as attention to irrelevant context indicates unreliable reasoning. Recent studies attempt to characterize how tokens are processed within the model by analyzing internal representations, such as attention matrices (Sriramanan et al., 2024; Zhang et al., 2023) and hidden states (Chen et al., 2024a; Perez-Beltrachini & Lapata, 2025), and then use these characterizations to estimate response reliability. However, these white-box methods consider only a snapshot of the model's computation procedure, for example, from a single layer or via max pooling across layers. As a result, they lose the progressive propagation from the input text through successive layers to the final output, failing to accurately quantify how each context token is transformed and accumulated into the final prediction. This limitation renders these approaches insufficient for estimating prediction uncertainty in RAG.

We leverage the **information flow** (Ferrando & Voita, 2024) to establish a sequential layer-wise trace, which quantifies the importance of context tokens to the generated response. In particular, we propose two quantities:

1. **Emergence Order** captures the order in which input tokens are added into the principal information flow, which is extracted from the complete flow in Algorithm 1. This order highlights the relative importance of input tokens in response generation (Figure 1 (b)).

2. **Contribution Layout** characterizes each input token's contribution, defined as the aggregation of all admissible paths from each input token to the final-layer embedding of the last input token. This layout offers a holistic view of how input information propagates through the language model (Figure 1 (c)).

[1]Artificial Intelligence Thrust, The Hong Kong University of Science and Technology (Guangzhou) [2]Jilin University. Correspondence to: Sihong Xie <sihongxie@hkust-gz.edu.cn>.

*Proceedings of the 43rd International Conference on Machine Learning*, Seoul, South Korea. PMLR 306, 2026. Copyright 2026 by the author(s).

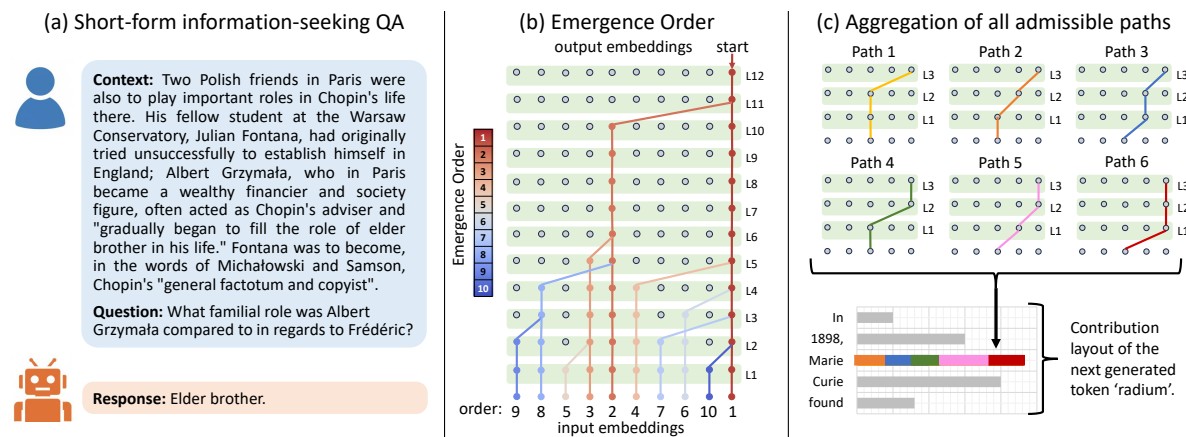

*Figure 1.* (a) An example of a short-form, information-seeking QA. (b) Principal information flow is extracted in reverse from the model's complete information flow, as detailed in Algorithm 1. The resulting Emergence Order records the sequence of input tokens added to this principal flow, with earlier tokens indicating greater importance for the final generation. For clarity, we neglect MLP operations as they operate independently on each token. (c) The contribution of each input token to the next generated token is defined as the sum of all valid paths from itself to the last input token embedding in the final layer. Contribution Layout represents the contributions of all input tokens.

We further leverage the two proposed quantities to estimate prediction reliability. Specifically, we introduce **simulatability**, which measures how well these quantities align with the true relevance of context tokens; higher alignment indicates greater response reliability. Additionally, since the contribution layout can be interpreted as a probability distribution, we quantify **concentration** by comparing it to a uniform distribution over the context tokens using KL divergence. This captures the extent to which contributions are focused on a small subset of tokens, with higher concentration reflecting increased model confidence. Finally, we use the results of these comparisons to optimize a calibrator, enabling more accurate estimation of response reliability.

The proposed method is empirically evaluated on short-form information-seeking question answering (QA) tasks (Figure 1 (a)). Experiments validate that our method outperforms existing baselines, with an average AUROC of 0.709, exceeding the runner-up performance of 0.676.

## 2. Related work

Prior research on uncertainty quantification (UQ) in LLMs that does not examine model mechanisms can be broadly categorized into two classes. Single-round generation methods primarily rely on token-level predictive probabilities derived from the softmax function (Margatina et al., 2023; Manakul et al., 2023; Fadeeva et al., 2024), entropy computed over the model's output vocabulary (Duan et al., 2024), or by querying the LLM itself to verify the correctness of its output (Kadavath et al., 2022). In contrast, multi-round generation methods estimate uncertainty by evaluating either the semantic consistency (Kuhn et al., 2023; Lin et al., 2023) or the entropy (Malinin & Gales, 2021) across multiple responses produced from a single input. These approaches

depend on the assumption that reliable predictions should remain stable under different sampling conditions. In addition, conformal prediction relies on the model's performance on a held-out calibration set, together with the i.i.d. assumption, to derive distribution-free uncertainty guarantees (Kumar et al., 2023; Quach et al., 2024).

Yet, estimating uncertainty solely at the model's output space is inadequate in RAG settings (Yu et al., 2024; Xu et al., 2024; Wang et al., 2024), because response reliability strongly depends on how the model processes the retrieved evidence. This motivates a shift in UQ from measuring output likelihood to evaluating whether predictions are properly grounded in query-relevant context, requiring a deep understanding of language model mechanisms. Recent UQ works start to unpack models. For example, Sriramanan et al. (2024) conduct eigen analysis on attention matrices and hidden states on a single chosen layer, and Zhang et al. (2023) compute token importance by max-pooling attention matrices over multiple layers. However, these methods collapse the sequential layer-wise computation into a single statistic. As a result, they discard the critical ordering and progressive transformation of information through the network, thereby obscuring the causal chain of how context tokens influence the final prediction.

## 3. Background

The computations of multi-head attention[1] within each layer can be reformulated as a direct expression of the input representations (Kobayashi et al., 2021). Consider a sequence of token embeddings $\mathbf{X} = [\mathbf{x}_1, ....., \mathbf{x}_T] = [\mathbf{x}_i]_{i=1}^T \in \mathbb{R}^{d \times T}$,

---

[1]We omit the MLP and layer normalization operations here because they operate independently on each token and do not affect the inter-token interactions we focus on.

where each embedding is $\mathbf{x}_i = \mathbf{X}_{:,i} \in \mathbb{R}^d$. The language model has $H$ heads, each of dimension $d_H = d/H$. For each head $h$, the corresponding query, key, and value projection matrices are denoted as $\mathbf{W}_Q^h$, $\mathbf{W}_K^h$, and $\mathbf{W}_V^h \in \mathbb{R}^{d_H \times d}$.

The query, key, and value vectors for embedding $\mathbf{x}_i$ are

$$\mathbf{q}_i^h = \mathbf{W}_Q^h \mathbf{x}_i, \quad \mathbf{k}_i^h = \mathbf{W}_K^h \mathbf{x}_i, \quad \mathbf{v}_i^h = \mathbf{W}_V^h \mathbf{x}_i. \quad (1)$$

The attention between $\mathbf{x}_i$ and $\mathbf{x}_j$ in head $h$ is defined as

$$\mathbf{A}_{i,j}^h = \frac{\exp(\langle \mathbf{q}_i^h, \mathbf{k}_j^h \rangle / \sqrt{d_H} + \mathbf{M}_{i,j})}{\sum_{t=1}^T \exp(\langle \mathbf{q}_i^h, \mathbf{k}_t^h \rangle / \sqrt{d_H} + \mathbf{M}_{i,t})} \in \mathbb{R}, \quad (2)$$

where $\mathbf{M} \in \mathbb{R}^{T \times T}$ is a causal mask such that $\mathbf{M}_{i,j} = -\infty$ if $j > i$ otherwise 0.

Accordingly, the output of head $h$ for input $\mathbf{x}_i$ is $\mathbf{z}_i^h = \sum_{j=1}^T \mathbf{A}_{i,j}^h \mathbf{v}_j^h \in \mathbb{R}^{d_H}$. Concatenating the outputs of all heads and projecting through $\mathbf{W}_O \in \mathbb{R}^{d \times d}$ yields $\mathbf{W}_O \cdot \text{Concat}(\mathbf{z}_i^1, ..., \mathbf{z}_i^H) \in \mathbb{R}^d$. Equivalently, partitioning $\mathbf{W}_O$ into submatrices $\mathbf{W}_O^h \in \mathbb{R}^{d \times d_H}$ allows to express the projection as a summation. Finally, incorporating the residual connection gives

$$\mathbf{y}_i = \mathbf{x}_i + \sum_{h=1}^H \mathbf{W}_O^h \mathbf{z}_i^h \in \mathbb{R}^d. \quad (3)$$

This allows the attribution vector from the input embedding $\mathbf{x}_j$ to the output embedding $\mathbf{y}_i$ to be defined by

$$\mathbf{a}_{j \to i} = \mathbf{1}_{\{j=i\}} \mathbf{x}_i + \sum_{h=1}^H \mathbf{W}_O^h \mathbf{A}_{i,j}^h \mathbf{W}_V^h \mathbf{x}_j \in \mathbb{R}^d,$$

where $\mathbf{1}_{\{j=i\}}$ equals 1 if $j = i$ and 0 otherwise. The contribution of $\mathbf{x}_j$ to $\mathbf{y}_i$ is quantified via the Manhattan distance:

$$\text{contri}(\mathbf{y}_i, \mathbf{a}_{j \to i})$$
$$= \frac{\max(0, ||\mathbf{y}_i||_1 - ||\mathbf{y}_i - \mathbf{a}_{j \to i}||_1)}{\sum_{t=1}^T \max(0, ||\mathbf{y}_i||_1 - ||\mathbf{y}_i - \mathbf{a}_{t \to i}||_1)} \in \mathbb{R}. \quad (4)$$

Intuitively, if $\mathbf{y}_i$ is close to its attribution-removed counterpart $\mathbf{y}_i - \mathbf{a}_{j \to i}$, then the contribution of $\mathbf{x}_j$ to $\mathbf{y}_i$ is small, as visualized in Figure 2 (Ferrando et al., 2022).

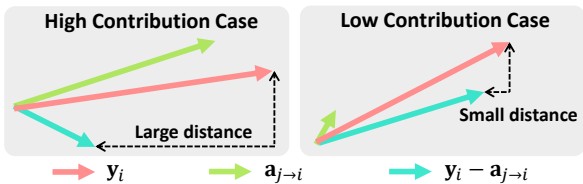

*Figure 2.* Manhattan-distance–based contribution of $\mathbf{x}_j$ to $\mathbf{y}_i$ .

Based on Eq. (4), we build a contribution matrix $\mathbf{C} \in \mathbb{R}^{T \times T}$ where the $(i, j)$-th entry $\mathbf{C}_{i,j} = \text{contri}(\mathbf{y}_i, \mathbf{a}_{j \to i})$. Since attention in causal language models is autoregressive, each token can only depend on itself and its predecessors. Hence, $\mathbf{A}_{i,j}^h = 0$ if $j > i$. As a result, $\mathbf{C}$ is lower-triangular with all

entries above the main diagonal equal to zero. For a model with $L$ transformer layers, we compute a contribution matrix at each layer, denoted $\mathbf{C}^{(l)}$ for $l = 1, \ldots, L$. The input and output embeddings of the $i$-th token at layer $l$ are denoted by $\mathbf{x}_i^{(l)}$ and $\mathbf{y}_i^{(l)}$, respectively. The collection of these layer-wise matrices, $\{\mathbf{C}^{(l)}\}_{l=1}^L$, is the complete information flow through the model. Figure 3 visualizes the matrices, where a consistent color is used for elements and flows targeting the same token.

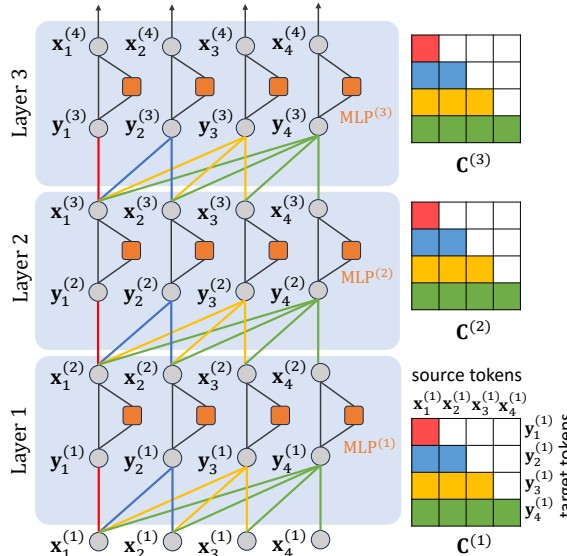

*Figure 3.* Layer-wise contribution matrices.

## 4. Method

Using the layer-wise contribution matrices $\{\mathbf{C}^{(l)}\}_{l=1}^L$, we define emergence order and contribution layout to measure the relative importance of input tokens to the output.

### 4.1. Emergence order

As the model relies solely on the final-layer representation of the last input token to predict the next token, Ferrando and Voita (2024) start from this representation to extract a subflow from $\{\mathbf{C}^{(l)}\}_{l=1}^L$ in a backward manner, using a pre-specified threshold. Input tokens included in the subflow are considered important for the generation. However, this binary criterion merely separates tokens into 'high-contribution' and 'low-contribution' classes based on a fixed threshold with continuous measure of relative importance.

To this end, we introduce the **Auto-Emergence** algorithm. This algorithm extracts principal information flows, denoted as $\{\mathbf{P}^{(l)}\}_{l=1}^L$, from the complete layer-wise contribution matrices $\{\mathbf{C}^{(l)}\}_{l=1}^L$. The process produces a vector $\mathbf{E} \in \mathbb{R}^T$ that records the **Emergence Order** of input tokens, where a token's earlier position reflects its greater relative significance to the generated output.

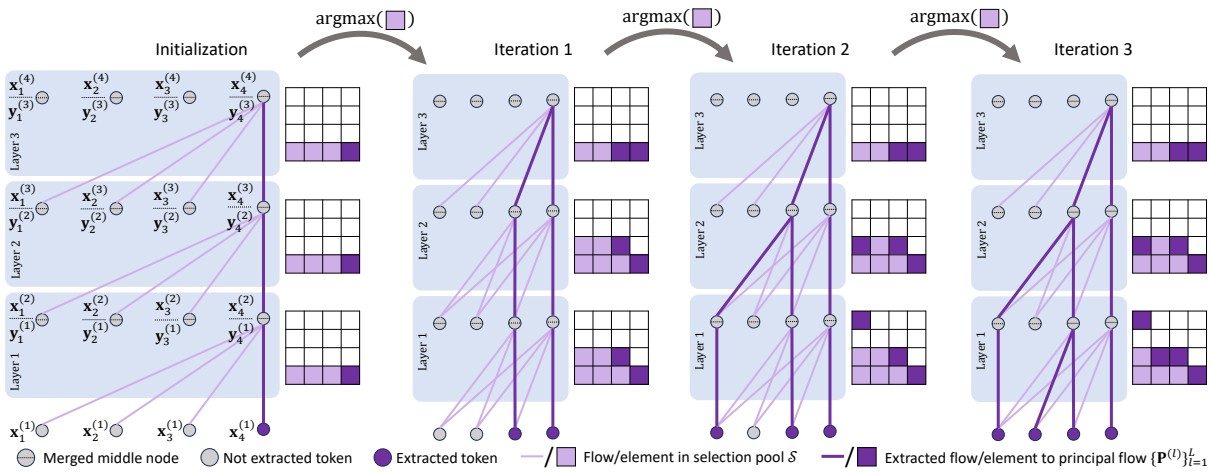

*Figure 4.* Demonstration of the Auto-Emergence algorithm. We extract the principal information flow $\{\mathbf{P}^{(l)}\}_{l=1}^{L}$ from the layer-wise contribution matrices $\{\mathbf{C}^{(l)}\}_{l=1}^{L}$ by iteratively selecting the strongest flow from the selection pool $\mathcal{S}$. The emergence order of the example is $\mathbf{E} = [3, 4, 2, 1]$. MLPs operating independently on each token are merged within the middle nodes to simplify the representation.

The algorithm begins at the final layer $L$ with the last input token's output embedding $\mathbf{y}_T^{(L)}$, whose self-contributions $\mathbf{C}_{T,T}^{(l)}$ are extracted into the corresponding principal flow element $\mathbf{P}_{T,T}^{(l)}$ for $l = 1, ..., L$, since self-contributions are typically dominant due to residual connections, as explained in Appendix A. We then assign $\mathbf{E}_T = 1$ and create a selection pool $\mathcal{S}$, which consists of all flows connected to the extracted ones. Subsequent extraction is an iterative top-down search. At each step, we extract the strongest flow from the pool $\mathcal{S}$. When a flow $\mathbf{C}_{i,j}^{(k)}$ is incorporated, its self-contributions from preceding layers, $\mathbf{C}_{j,j}^{(l)}$ for $l < k$, are also extracted. $\mathbf{E}_j$ is then assigned the next available rank and $\mathcal{S}$ will be updated accordingly. Once a non-zero rank has been assigned to $\mathbf{E}_j$, it remains fixed in subsequent iterations. This is enforced by removing the corresponding $\mathbf{C}_{i,j}^{(k)}$ and $\mathbf{C}_{j,j}^{(l)}$ for $l < k$ from the pool $\mathcal{S}$. The process continues until all tokens are ranked. The vector $\mathbf{E}$ reveals how the input tokens influence the generation process. The method is illustrated with an example in Figure 4, and is detailed in Algorithm 1.

### 4.2. Contribution layout

The collection of complete layer-wise contribution matrices $\{\mathbf{C}^{(l)}\}_{l=1}^{L}$ characterizes local information flow within each transformer layer. However, these matrices are high-dimensional and difficult to interpret. To obtain a compact description of how input tokens influence the final-layer representations, we introduce a total contribution matrix:

$$\mathbf{C}^{\text{total}} = \mathbf{C}^{(L)}\mathbf{C}^{(L-1)} \cdots \mathbf{C}^{(1)} \in \mathbb{R}^{T \times T}. \quad (5)$$

The entry $\mathbf{C}_{i,j}^{\text{total}}$ measures the overall contribution from the source token embedding $\mathbf{x}_j^{(1)}$ at the input layer to the target

---

**Algorithm 1** Auto-Emergence Algorithm

**Require:** contribution matrices $\{\mathbf{C}^{(l)}\}_{l=1}^{L}$

---

**Initialization:**
$\quad \mathbf{P}^{(l)} \leftarrow 0 \in \mathbb{R}^{T \times T} \quad \forall l = 1, \ldots, L$
$\quad \mathbf{E} \leftarrow 0 \in \mathbb{R}^{T}$
$\quad \mathbf{E}_T \leftarrow 1$
$\quad \mathbf{P}_{T,T}^{(l)} \leftarrow \mathbf{C}_{T,T}^{(l)} \quad \forall l = 1, \ldots, L$
$\quad \mathcal{S} \leftarrow \{\mathbf{C}_{T,j}^{(l)} \mid j < T, l \leq L\}$

---

**Extraction:**
**while** $\exists i$ s.t. $\mathbf{E}_i = 0$ **do**
$\quad$ Select $\mathbf{C}_{i,j}^{(k)} = \arg\max(\mathcal{S})$
$\quad \mathbf{P}_{i,j}^{(k)} \leftarrow \mathbf{C}_{i,j}^{(k)}$
$\quad \mathbf{P}_{j,j}^{(l)} \leftarrow \mathbf{C}_{j,j}^{(l)}$ for $l = 1, \ldots, k-1$
$\quad \mathcal{S} \leftarrow \mathcal{S} \setminus (\{\mathbf{C}_{i,j}^{(k)}\} \cup \{\mathbf{C}_{j,j}^{(l)} \mid l = 1, \ldots, k-1\})$
$\quad \mathbf{E}_j \leftarrow \min\{r \in \mathbb{Z}^+ \mid r \notin \mathbf{E}\}$
$\quad \mathcal{S} \leftarrow \mathcal{S} \cup \{\mathbf{C}_{j,m}^{(l)} \mid m < j, l \leq k-1\}$
**end while**
**return** $\{\mathbf{P}^{(l)}\}_{l=1}^{L}, \mathbf{E}$

---

token embedding $\mathbf{y}_i^{(L)}$ at the final layer. In particular, it is a sum over all valid paths that connect the $j$-th token at the input layer to the $i$-th token at the final layer through

$$\mathbf{C}_{i,j}^{\text{total}} = \sum_{s_{L-1}=j}^{i} \cdots \sum_{s_2=j}^{s_3} \sum_{s_1=j}^{s_2} \mathbf{C}_{i,s_{L-1}}^{(L)} \cdots \mathbf{C}_{s_2,s_1}^{(2)} \mathbf{C}_{s_1,j}^{(1)}.$$

The summation indices $(s_1, \ldots, s_{L-1})$ enumerate all admissible intermediate tokens along the path from source $j$ to target $i$, implicitly subject to the monotonicity constraint

$$j \leq s_1 \leq s_2 \leq \cdots \leq s_{L-1} \leq i. \quad (6)$$

This ensures information flows consistently "upstream," never skipping or exceeding the valid range between $i$ and $j$. The constraint follows from causal masking in autoregressive transformers, where each token can only attend to preceding tokens and influence subsequent ones. Under this constraint, each valid index sequence defines an admissible path from $j$ to $i$, whose strength is given by the product of per-layer contributions along the path.

For example in Figure 5, for a two-layer model ($L = 2$), the total contribution from input $\mathbf{x}_1^{(1)}$ to the final-layer output $\mathbf{y}_3^{(2)}$ can be expressed as

$$\mathbf{C}_{3,1}^{\text{total}} = \sum_{s_1=1}^{3} \mathbf{C}_{3,s_1}^{(2)} \mathbf{C}_{s_1,1}^{(1)}$$
$$= \mathbf{C}_{3,1}^{(2)}\mathbf{C}_{1,1}^{(1)} + \mathbf{C}_{3,2}^{(2)}\mathbf{C}_{2,1}^{(1)} + \mathbf{C}_{3,3}^{(2)}\mathbf{C}_{3,1}^{(1)},$$

where each term corresponds to a distinct path via $\mathbf{y}_1^{(1)}$, $\mathbf{y}_2^{(1)}$, and $\mathbf{y}_3^{(1)}$, respectively. Thus, $\mathbf{C}^{\text{total}}$ consolidates the entire information flow into a single interpretable matrix, while preserving the underlying path semantics across layers. Since only the last token's output representation is used to generate the next token, the **Contribution Layout** is given by the last row of $\mathbf{C}^{\text{total}}$, capturing the influence of all input tokens on the generated token:

$$\mathbf{C}^{\text{layout}} = \mathbf{C}_{-1,:}^{\text{total}} \in \mathbb{R}^T. \tag{7}$$

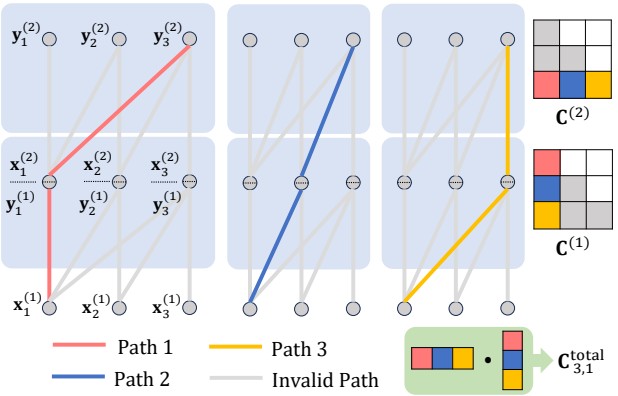

*Figure 5.* Admissible paths from the first to the third token in a two-layer transformer are highlighted in different colors. Their aggregation represents the total contribution of the first token.

While $\mathbf{C}^{\text{layout}}$ captures the influence of all input tokens on the generated token, it often contains small, noisy flows from less relevant paths. To address this, we leverage the principal flows $\{\mathbf{P}^{(l)}\}_{l=1}^{L}$ obtained from Algorithm 1 to compute a principal contribution layout:

$$\mathbf{P}^{\text{total}} = \mathbf{P}^{(L)}\mathbf{P}^{(L-1)}\cdots\mathbf{P}^{(1)} \in \mathbb{R}^{T \times T},$$
$$\mathbf{P}^{\text{layout}} = \mathbf{P}_{-1,:}^{\text{total}} \in \mathbb{R}^T, \tag{8}$$

which provides a more interpretable representation of the dominant contributions to generation.

## 4.3. Complementary views of information flow

Emergence order, $\mathbf{E}$, and contribution layout, $\mathbf{C}^{\text{layout}}$, capture two complementary perspectives on how information flows from context tokens to a model's prediction.

Emergence order is computed by tracing backward from the final-layer decisive representation of the last input token. Tokens that appear earlier in this backward traversal are those whose information is incorporated more directly and prominently into the final decision, and can therefore be interpreted as primary evidence for the model's output.

Contribution layout, in contrast, aggregates influence over all admissible paths through the network, including indirect and multi-hop interactions mediated by intermediate tokens and layers. This allows it to capture tokens that may not serve as direct evidence for the prediction but are nonetheless essential in building a comprehensive understanding of the context. Their influence accumulates across the network, making the layout more distributed.

For example, consider the sentence:

```
"Alice gave Bob the book after she
finished reading it",
```

where the task is to resolve `"she"`, and the model correctly predicts `"Alice"`. Emergence order highlights direct, decision-critical tokens such as `"she"` and `"Alice"` in the context, whereas contribution layout also assigns importance to tokens like `"book"` and `"reading"`, which help establish the semantic context.

To provide intuition, we draw an analogy to a traffic network, where tokens correspond to cities, and the model functions as a transportation system routing information toward the final prediction. Emergence order identifies which cities' traffic emerge earlier when traced backward from the destination, highlighting sources that directly drive the final outcome. In contrast, contribution layout accounts for all routes through which traffic from each city reaches the destination, including multi-step pathways through intermediate nodes. This distinction reflects that emergence order captures direct, decision-critical influence, whereas contribution layout additionally incorporates distributed, context-building effects across the network.

In addition, in Eq. (8), the principal contribution layout, $\mathbf{P}^{\text{layout}}$, integrates both perspectives, identifying context tokens that serve simultaneously as direct evidence and as facilitators of semantic context.

## 4.4. Context slicing across multiple generations

In RAG, the input sequence typically consists of an instruction prompt, a context, and a question, in that order. Let the total number of input tokens be $T = T_p + T_c + T_q$, where $T_p$, $T_c$, and $T_q$ denote the lengths of the instruction prompt, context, and question, respectively. The emergence order $\mathbf{E}$ and the contribution layouts $\mathbf{C}^{\text{layout}}$, $\mathbf{P}^{\text{layout}}$ are defined over the full sequence of $T$ input tokens. Since our primary interest lies in how LLMs process the retrieved context, we focus on the context segment of the input sequence. Formally, we define the index list $\mathbf{I} = [T_p + 1, ..., T_p + T_c]$, which corresponds to the token positions allocated to the context. For a single generated token, the sliced emergence order and contribution layouts are then defined as

$$\mathbf{E_I} := [\mathbf{E}_i]_{i \in \mathbf{I}} \in \mathbb{R}^{T_c}, \quad \mathbf{C_I^{\text{layout}}} := [\mathbf{C}_i^{\text{layout}}]_{i \in \mathbf{I}} \in \mathbb{R}^{T_c},$$
$$\mathbf{P_I^{\text{layout}}} := [\mathbf{P}_i^{\text{layout}}]_{i \in \mathbf{I}} \in \mathbb{R}^{T_c}.$$

When generating multiple tokens, we obtain a sequence of context emergence order $\{\mathbf{E_I}(t)\}_{t=1}^{T_g}$ and contribution layouts $\{\mathbf{C_I^{\text{layout}}}(t)\}_{t=1}^{T_g}$, $\{\mathbf{P_I^{\text{layout}}}(t)\}_{t=1}^{T_g}$, where $T_g$ is the number of generated tokens. We then aggregate over all generated tokens to form averaged results in $\mathbb{R}^{T_c}$:

$$\bar{\mathbf{E}}_\mathbf{I} = \frac{1}{T_g} \sum_{t=1}^{T_g} \mathbf{E_I}(t), \quad \bar{\mathbf{C}}_\mathbf{I}^{\text{layout}} = \frac{1}{T_g} \sum_{t=1}^{T_g} \mathbf{C_I^{\text{layout}}}(t),$$
$$\bar{\mathbf{P}}_\mathbf{I}^{\text{layout}} = \frac{1}{T_g} \sum_{t=1}^{T_g} \mathbf{P_I^{\text{layout}}}(t).$$

## 4.5. Calibrated confidence of LLM responses in RAG

### 4.5.1. RELEVANCE LAYOUT

To establish an evaluation reference, we use a ranker to assign a scalar score $r$ representing the overall relevance of a context to a given question. While this score reflects the model's holistic judgment, it does not reveal the role of individual context tokens in making the decision. To decompose this relevance score into token-level signals, we make use of Shapley values, a well-established concept from cooperative game theory (Lundberg & Lee, 2017; Sundararajan & Najmi, 2020). The key intuition is to treat each token in the context as a "player" in a cooperative game, where the total payoff is the relevance score $r$. The Shapley framework then assigns each token a fair share of this payoff by averaging its marginal effect across all possible subsets of tokens. From these token-level attributions, we construct the **Relevance Layout**, denoted by $\mathbf{R}^{\text{layout}} \in \mathbb{R}^{T_c}$, to provide a fine-grained view of how the ranker assesses the usefulness of each token for answering the question. It serves as an estimated ground truth against which we compare the model-derived results in our UQ framework.

### 4.5.2. FIDELITY OF MODEL-DERIVED LAYOUTS

**Simulatability.** We introduce simulatability as a measure of how well the model's internal context processing aligns with an external notion of token relevance. Intuitively, if the emergence order $\bar{\mathbf{E}}_\mathbf{I}$ and the contribution layouts $\bar{\mathbf{C}}_\mathbf{I}^{\text{layout}}$ and $\bar{\mathbf{P}}_\mathbf{I}^{\text{layout}}$ place emphasis on the same tokens as the estimated relevance layout $\mathbf{R}^{\text{layout}}$, then the model's reasoning process and response are more reliable.

The comparison starts by ranking context tokens. Since $\mathbf{R}^{\text{layout}}$, $\bar{\mathbf{C}}_\mathbf{I}^{\text{layout}}$, and $\bar{\mathbf{P}}_\mathbf{I}^{\text{layout}}$ assign a real-valued score to each context token, with larger values indicating higher importance, we transform them into index lists by sorting their values in descending order. In contrast, $\bar{\mathbf{E}}_\mathbf{I}$ encodes an emergence order: smaller values indicate earlier entry into the principal flow, and thus greater importance. Accordingly, we sort its indices in ascending order. Formally, denoting $\pi$ a permutation of context token indices, we have

$$\pi_\mathbf{R} = \text{argsort}_\downarrow(\mathbf{R}^{\text{layout}}), \quad \pi_\mathbf{C} = \text{argsort}_\downarrow(\bar{\mathbf{C}}_\mathbf{I}^{\text{layout}}),$$
$$\pi_\mathbf{P} = \text{argsort}_\downarrow(\bar{\mathbf{P}}_\mathbf{I}^{\text{layout}}), \quad \pi_\mathbf{E} = \text{argsort}_\uparrow(\bar{\mathbf{E}}_\mathbf{I}).$$

We evaluate simulatability by comparing $\pi_\mathbf{C}$, $\pi_\mathbf{P}$, and $\pi_\mathbf{E}$ against $\pi_\mathbf{R}$ using **rank-biased overlap (RBO)** (Webber et al., 2010), which emphasizes agreement at higher-ranked tokens. Specifically, we compute $\text{RBO}(\pi_\mathbf{C}, \pi_\mathbf{R})$, $\text{RBO}(\pi_\mathbf{P}, \pi_\mathbf{R})$, and $\text{RBO}(\pi_\mathbf{E}, \pi_\mathbf{R})$ to quantify the alignment of each ranking with $\pi_\mathbf{R}$. The computation is governed by a persistence parameter $p$, with larger values giving more weight to lower-ranked items and smaller values emphasizing higher-ranked items. In our experiments, we set $p = 0.7$. Details of the RBO computation and the effects of different $p$ values are provided in Appendix B.

**Concentration.** Understanding how focused a model's reasoning is across context tokens can reveal whether the model relies on a few key context tokens or distributes contributions broadly. To quantify this, we examine the concentration of both $\bar{\mathbf{C}}_\mathbf{I}^{\text{layout}}$ and $\bar{\mathbf{P}}_\mathbf{I}^{\text{layout}}$. Highly concentrated layouts indicate that a small subset of tokens dominates the model's internal computation, reflecting strong confidence, whereas uniform layouts indicate evenly distributed contribution.

Since both $\bar{\mathbf{C}}_\mathbf{I}^{\text{layout}}$ and $\bar{\mathbf{P}}_\mathbf{I}^{\text{layout}}$ assign positive real-valued scores to each context token, denoting $\Delta^{T_c-1}$ the standard simplex in $\mathbb{R}^{T_c}$, we first normalize them to the simplex:

$$\hat{\mathbf{C}}_\mathbf{I}^{\text{layout}} = \frac{\bar{\mathbf{C}}_\mathbf{I}^{\text{layout}}}{\sum_{i \in \mathbf{I}} \bar{\mathbf{C}}_i^{\text{layout}}} \in \Delta^{T_c-1},$$
$$\hat{\mathbf{P}}_\mathbf{I}^{\text{layout}} = \frac{\bar{\mathbf{P}}_\mathbf{I}^{\text{layout}}}{\sum_{i \in \mathbf{I}} \bar{\mathbf{P}}_i^{\text{layout}}} \in \Delta^{T_c-1}.$$

The uniform distribution over $T_c$ tokens is defined as

$$\mathbf{U} = \left[ \frac{1}{T_c}, \frac{1}{T_c}, \ldots, \frac{1}{T_c} \right] \in \Delta^{T_c - 1}.$$

We measure the deviation of the layouts from uniformity using Kullback–Leibler (KL) divergence (Van Erven & Harremos, 2014), denoted as $\mathrm{KL}(\widehat{\mathbf{C}}_{\mathbf{I}}^{\text{layout}} | \mathbf{U})$ and $\mathrm{KL}(\widehat{\mathbf{P}}_{\mathbf{I}}^{\text{layout}} | \mathbf{U})$. This quantifies whether the layouts are concentrated on a small subset of tokens. The details of the discrete KL computation are provided in Appendix C.

### 4.5.3. CALIBRATOR FOR RESPONSE CONFIDENCE

We develop a multi-level granularity that groups context tokens into word- and phrase-level units for computing emergence order and contribution layouts, which are then used to measure simulatability (see Appendix D for details). In addition, we use the scalar score $r$ from the ranker as an indicator of overall context relevance: higher values of $r$ indicate that the context is more pertinent to the question, increasing the reliability of the generated response.

Finally, we combine these features—simulatability, concentration, and the context relevance score $r$—to train a calibrator that outputs a calibrated confidence for the model's response. Together, these features enable the calibrator to provide confidence estimates that reflect both the model's internal reasoning and the quality of the retrieved context.

## 5. Experiment

### 5.1. Experimental setup

**Datasets** We evaluate our method on SQuAD2.0 (Rajpurkar et al., 2018) and MS MARCO (Nguyen et al., 2016), both of which include distractor passages that naturally reflect the noise of real-world retrieval systems. We further evaluate on HotpotQA (Yang et al., 2018) to assess performance on complex multi-hop reasoning tasks, including bridge- and comparison-type questions that require synthesizing multiple passages. Each dataset are split into training, validation, and test sets with a 3:1:1 ratio.

**Models** We use LLaMA-3.2-3B-Instruct, LLaMA-3-8B-Instruct (AI@Meta, 2024) and Gemma-3-4B-it (Google, 2025) as the base question answering (QA) models. Qwen-3-Reranker-8B (Zhang et al., 2025) is used as the ranker to assign a relevance score to each context. The correctness of predictions is evaluated with HHEM-2.1-Open (Bao et al., 2024). The calibrator introduced in Section 4.5.3 is trained with XGBoost library (Chen & Guestrin, 2016). Hyperparameter optimization is performed on the validation set using Optuna (Akiba et al., 2019). The inference prompt and the detailed correctness evaluation is described in Appendix E. The source code is hosted at `https://github.com/rxu0112/RAG-information-flow`.

**Baselines.** We evaluate our approach against extensive baseline methods, falling into the following categories.

- **Likelihood-based methods**: Perplexity (PPL) (Margatina et al., 2023) quantifies uncertainty by measuring predicted softmax probabilities of the output tokens.
- **Consistency-based methods**: Regular Entropy (Malinin & Gales, 2021) averages the predictive entropy of sampled outputs, and Semantic Entropy (Kuhn et al., 2023) measures consistency in their semantic content.
- **Attention-based methods**: Attention Score (Sriramanan et al., 2024) estimates uncertainty using the eigenvalues of attention matrices, while Focus (Zhang et al., 2023) leverages attention matrices to quantify token importance for uncertainty estimation.
- **Embedding-based methods**: KnowingMore (Zhang et al., 2021) integrates question and context embeddings to calibrate prediction confidence. Utility Ranker (Perez-Beltrachini & Lapata, 2025) uses embeddings to estimate the usefulness of retrieved context for the question.
- **Verbalized methods**: P(True) (Kadavath et al., 2022) queries the QA model for its self-assessed confidence.

### 5.2. Results

As shown in Table 1, our proposed method effectively outperforms existing uncertainty quantification approaches. While it does not dominate every individual metric, it achieves the best overall mean rank for the three language models, indicating strong and stable performance. In particular, our method attains the highest AUROC and AUPRC on SQuAD2.0 and MS MARCO, and remains competitive on HotpotQA. These results demonstrate that modeling progressive information flow yields more reliable uncertainty estimates. We further examine the discriminative ability of the proposed simulatability and concentration with respect to the relevance scores $r$ in Appendix G. Since P(True), KnowingMore, and our method produce calibrated confidence scores, we evaluate their calibration performance in the Appendix I. Finally, an analysis of the computational cost and applicability is also provided in Appendix H.

## 6. Discussion

### 6.1. Out-of-distribution generalizability

A key advantage of our information-flow-based metrics is their enhanced interpretability, as they are grounded in the alignment between token contributions and their true relevance, rather than relying on superficial dataset patterns. We posit that this makes our approach inherently more robust to distribution shifts. We validate this claim by applying calibrators trained on one dataset to test data from entirely different distributions in Table 2. Crucially, we observe that different calibrators perform similarly on a given test

*Table 1.* Uncertainty quantification performance of different methods on three datasets, evaluated with two base QA models using AUROC and AUPRC. **Bold** denotes the best value for each metric, while the runner-up is underlined. The Mean Rank is computed by averaging the per-dataset ranks of each method across all datasets, with lower values indicating better performance. Only Semantic Entropy, Utility Ranker, and our method involve randomness; we report the standard deviation of their results in Appendix F.

| Model | Method | SQuAD2.0 | | HotpotQA | | MS MARCO | | Mean Rank |
|---|---|---|---|---|---|---|---|---|
| | | AUROC | AUPRC | AUROC | AUPRC | AUROC | AUPRC | |
| LLaMA-3.2 3B-Instruct | PPL | 0.608 | 0.794 | 0.582 | 0.912 | 0.592 | 0.691 | 5.67 |
| | P(True) | 0.547 | 0.746 | 0.467 | 0.873 | 0.557 | 0.664 | 7.67 |
| | Regular Entropy | 0.698 | 0.829 | 0.651 | 0.925 | 0.654 | 0.729 | 3.00 |
| | Semantic Entropy | 0.645 | 0.789 | 0.614 | 0.911 | 0.528 | 0.611 | 6.08 |
| | Attention Score | 0.513 | 0.718 | 0.478 | 0.867 | 0.459 | 0.561 | 8.83 |
| | Focus | 0.703 | 0.830 | **0.701** | **0.944** | 0.690 | 0.752 | 1.67 |
| | KnowingMore | 0.663 | 0.827 | 0.590 | 0.905 | 0.623 | 0.702 | 4.75 |
| | Utility Ranker | 0.629 | 0.789 | 0.597 | 0.905 | 0.593 | 0.661 | 6.00 |
| | Ours | **0.735** | **0.855** | 0.672 | 0.934 | **0.727** | **0.778** | **1.33** |
| LLaMA-3 8B-Instruct | PPL | 0.600 | 0.636 | 0.531 | 0.857 | 0.529 | 0.605 | 6.67 |
| | P(True) | 0.676 | 0.687 | 0.622 | 0.886 | 0.554 | 0.626 | 3.83 |
| | Regular Entropy | 0.649 | 0.656 | 0.583 | 0.873 | 0.568 | 0.624 | 5.00 |
| | Semantic Entropy | 0.515 | 0.542 | 0.493 | 0.836 | 0.518 | 0.582 | 8.00 |
| | Attention Score | 0.493 | 0.535 | 0.492 | 0.831 | 0.485 | 0.559 | 9.00 |
| | Focus | 0.719 | 0.724 | 0.703 | 0.916 | **0.693** | 0.726 | 1.83 |
| | KnowingMore | 0.637 | 0.668 | 0.593 | 0.876 | 0.608 | 0.677 | 3.83 |
| | Utility Ranker | 0.569 | 0.588 | 0.561 | 0.867 | 0.605 | 0.650 | 5.67 |
| | Ours | **0.778** | **0.782** | **0.727** | **0.927** | 0.690 | **0.737** | **1.17** |
| Gemma-3 4B-it | PPL | 0.639 | 0.622 | 0.605 | 0.772 | 0.561 | 0.495 | 5.00 |
| | P(True) | 0.545 | 0.521 | 0.525 | 0.725 | 0.548 | 0.484 | 7.83 |
| | Regular Entropy | 0.658 | 0.633 | 0.617 | 0.779 | 0.570 | 0.505 | 3.33 |
| | Semantic Entropy | 0.590 | 0.546 | 0.530 | 0.727 | 0.574 | 0.479 | 6.58 |
| | Attention Score | 0.529 | 0.518 | 0.477 | 0.698 | 0.473 | 0.413 | 9.00 |
| | Focus | 0.653 | 0.636 | **0.650** | **0.832** | 0.574 | 0.519 | 2.25 |
| | KnowingMore | 0.620 | 0.625 | 0.532 | 0.739 | 0.598 | 0.523 | 4.33 |
| | Utility Ranker | 0.643 | 0.614 | 0.545 | 0.744 | 0.564 | 0.486 | 5.33 |
| | Ours | **0.703** | **0.684** | 0.645 | 0.814 | **0.706** | **0.627** | **1.33** |

set. The consistency across training sources provides strong evidence that our method generalizes well, mitigating the problem of sensitivity to train-test distribution shifts.

### 6.2. Faithfulness demonstration

To move beyond correlation and establish the causal faithfulness of the identified information flows, we conducted a controlled ablation study. Specifically, we randomly selected 100 samples for which the base QA model generated correct answers, as determined by HHEM-2.1-Open. For each sample, we used the three proposed measures, namely $\overline{\mathbf{E}}_{\mathbf{I}}$, $\overline{\mathbf{C}}_{\mathbf{I}}^{\text{layout}}$, and $\overline{\mathbf{P}}_{\mathbf{I}}^{\text{layout}}$, to identify the most and least critical context tokens. We then performed two ablation procedures: one where we ablated the five top-ranked tokens, and another where we ablated the five bottom-ranked tokens. We let the QA models infer on the modified samples and measure the resulting changes in prediction accuracy to assess whether the identified information flows reflect the actual reliance on the context.

The results, detailed in Figure 6, demonstrate a consistent and sharp contrast. Ablating the top-ranked tokens causes a

majority (over 50%) of the previously correct predictions to become incorrect. Conversely, ablating the bottom-ranked tokens results in negligible performance degradation. This difference provides direct causal evidence that the tokens ranked highly by our method are functionally necessary for the model's correct reasoning. This confirms that we identify tokens that genuinely drive predictions, rather than those that merely correlate with correct outcomes.

### 6.3. Ranker bias

Simulatability relies on a token-level ground-truth relevance layout obtained by Shapley decomposition of a ranker's single relevance score, as introduced in Section 4.5. This introduces potential circularity and bias: if the ranker's bias differs from the bias of the QA model, simulatability may reward the model for mimicking the ranker rather than genuinely grounding its predictions in relevant context. Thus, we measure the extent of bias in the estimated relevance layouts produced by the ranker model (Qwen-3-Reranker-8B). In particular, for each dataset, we employ human verification to filter ranker-produced relevance layouts, retaining only

*Table 2.* Out-of-distribution generalizability of the proposed method across three datasets, evaluated with two base QA models using AUROC and AUPRC. The method is trained on one dataset and tested on all three.

| Model | Test / Train | SQuAD2.0 | | HotpotQA | | MS MARCO | |
|---|---|---|---|---|---|---|---|
| | | AUROC | AUPRC | AUROC | AUPRC | AUROC | AUPRC |
| LLaMA-3.2 3B-Instruct | SQuAD2.0 | 0.735 | 0.855 | 0.658 | 0.930 | 0.695 | 0.759 |
| | HotpotQA | 0.728 | 0.838 | 0.672 | 0.934 | 0.692 | 0.752 |
| | MS MARCO | 0.715 | 0.833 | 0.633 | 0.923 | 0.727 | 0.778 |
| LLaMA-3 8B-Instruct | SQuAD2.0 | 0.778 | 0.782 | 0.718 | 0.922 | 0.656 | 0.709 |
| | HotpotQA | 0.745 | 0.757 | 0.727 | 0.927 | 0.645 | 0.703 |
| | MS MARCO | 0.722 | 0.738 | 0.680 | 0.911 | 0.690 | 0.737 |
| Gemma-3 4B-it | SQuAD2.0 | 0.703 | 0.684 | 0.623 | 0.801 | 0.655 | 0.576 |
| | HotpotQA | 0.663 | 0.652 | 0.645 | 0.814 | 0.672 | 0.574 |
| | MS MARCO | 0.625 | 0.645 | 0.633 | 0.779 | 0.706 | 0.627 |

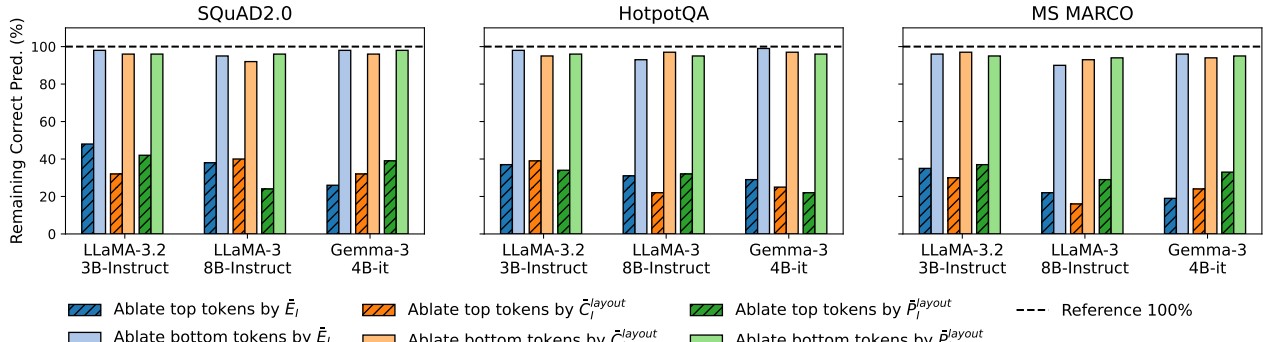

*Figure 6.* Fraction of correct predictions retained after ablating the top-5 and bottom-5 tokens, ranked by the three proposed measures.

those whose top-ranked tokens are genuinely relevant to the query. We then evaluate the simulatability performance on the filtered subset (see Appendix J for detailed results). We observe a modest increase in AUROC and AUPRC after human verification, indicating mild bias from the ranker. However, this is due to the lack of gold annotations rather than our information-flow method, while the limited magnitude of improvement suggests the ranker remains a practical choice for large-scale automatic evaluation.

Moreover, we assess our reliance on ranker selection by replacing Qwen-3-Reranker-8B with MSMARCO-MiniLM-L12-v2 (Reimers & Gurevych, 2019) and BGE-v2-m3 (Chen et al., 2024b), showing results in Appendix J.

### 6.4. Effect of Post-hoc Calibration

To ensure that the observed performance gains arise from our proposed metrics rather than the post-hoc calibration procedure, we applied the identical calibration pipeline to baseline methods. As shown in Appendix K, calibration does not improve baseline performance and frequently leads to degradation due to overfitting. In contrast, our method produces multiple complementary signals that must be aggregated into a single scalar uncertainty score for fair comparison under AUROC and AUPRC. Therefore, calibration in our framework primarily serves as an aggregation step.

## 7. Conclusion

In this work, we propose a novel UQ framework for retrieval-augmented LLMs that leverages the sequential layer-wise information flow to assess the importance of context tokens in generated responses. By introducing emergence order and contribution layout, along with simulatability and concentration, our method captures both the propagation and focus of contextual information within the model. Experiments validate that it provides more reliable uncertainty estimates than existing baselines, highlighting the value of incorporating the entire information propagation process.

## Impact Statement

This paper presents work whose goal is to advance the field of Machine Learning. There are many potential societal consequences of our work, none of which we feel must be specifically highlighted here.

## Acknowledgement

Sihong Xie was supported by the National Key R&D Program of China (Grant No.2023YFF0725001), the Department of Science and Technology of Guangdong Province (2023CX10X079), the Guangzhou-HKUST(GZ) Joint Funding Program (Grant No.2023A03J0008), and the Education Bureau Guangzhou Municipality.

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

## A. Dominance of self-contributions via residual connection

The dominance of self-contributions can be traced back to the architectural design of transformers. In particular, residual connections parameterize each layer as an additive update to its input, rather than a complete transformation. This formulation encourages the model to learn small, incremental refinements around an approximate identity mapping, so that information from earlier layers is largely preserved and progressively refined rather than overwritten (Jastrzbski et al., 2018).

Moreover, the use of pre-normalization further stabilizes this behavior. By normalizing inputs before each sub-layer, pre-normalization constrains the magnitude of layer-wise updates while allowing the residual stream itself to accumulate information across depth. Consequently, the residual pathway becomes the primary carrier of information, naturally amplifying the contribution of a token to its own representation over successive layers (Xiong et al., 2020).

This architectural bias aligns with our empirical observations. Even when residual connections are not explicitly prioritized, i.e., when they are included as candidates within the selection pool $S$ rather than directly injected into the principal information flow, they are consistently selected in the earliest iterations. This indicates that self-contributions dominate not as an artifact of our method, but as an inherent property of transformers induced by residual learning and pre-normalization.

## B. Computation of rank-biased overlap (RBO)

Rank-Biased Overlap (RBO) (Webber et al., 2010) is a measure of similarity between two ranked lists that ranges from 0 to 1 and emphasizes agreement at higher-ranked items. Values closer to 1 indicate that the two lists are highly similar, while values closer to 0 indicate that they are dissimilar.

Consider we have two permutations of the same indices of length $N$: $\pi_X = [x_1, x_2, \ldots, x_N]$ and $\pi_Y = [y_1, y_2, \ldots, y_N]$. At each depth $d = 1, \ldots, N$, compute the overlap between the top-$d$ items:

$$A_d = \{x_1, \ldots, x_d\} \cap \{y_1, \ldots, y_d\}, \quad \text{agr}(d) = |A_d|/d.$$

The RBO score applies a geometric weighting to emphasize higher ranks:

$$\text{RBO}_p(\pi_X, \pi_Y) = (1-p) \sum_{d=1}^{N} p^{d-1} \text{agr}(d), \quad 0 < p < 1,$$

where $p$ is the persistence parameter controlling the weight decay: larger $p$ assigns more weight to lower-ranked items, while smaller $p$ emphasizes top-ranked positions.

In our work, we apply RBO to compare the ranked index lists $\pi_E, \pi_C, \pi_P$ with the relevance ranking $\pi_R$, capturing the alignment between the model's internal processing and the estimated importance of context tokens. The results of our method in Table 1 are reported with $p = 0.7$.

To evaluate the impact of different RBO hyperparameters, we present the experiment results of $p$ in $\{0.1, 0.3, 0.5, 0.7, 0.9\}$ in Table 3. Varying the RBO hyperparameter does not significantly affect the performance of our method. This robustness arises because correct predictions exhibit strong agreement with the estimated relevance layout across a broad range of context positions, rather than being concentrated only among the top-ranked tokens. Consequently, changing $p$ does not substantially alter the relative ordering of examples (correct predictions consistently yield higher RBO scores), so RBO remains a reliable measure of prediction uncertainty across different $p$ values.

## C. KL Divergence for discrete probability distributions

Let $\mu = [\mu_1, \mu_2, \ldots, \mu_N]$ and $\nu = [\nu_1, \nu_2, \ldots, \nu_N]$ denote two discrete probability distributions over $N$ elements, with $\mu_i, \nu_i \geq 0$ and $\sum_{i=1}^{N} \mu_i = \sum_{i=1}^{N} \nu_i = 1$. The Kullback-Leibler (KL) divergence from $\mu$ to $\nu$ is defined as

$$\text{KL}(\mu \| \nu) = \sum_{i=1}^{N} \mu_i \log \frac{\mu_i}{\nu_i}.$$

In our setting, $\mu$ corresponds to the normalized contribution layouts $\widehat{\mathbf{C}}_{\mathbf{I}}^{\text{layout}}$ and $\widehat{\mathbf{P}}_{\mathbf{I}}^{\text{layout}}$ and $\nu$ corresponds to the uniform distribution over $T_c$ tokens $\mathbf{U} = \left[\frac{1}{T_c}, \frac{1}{T_c}, \ldots, \frac{1}{T_c}\right] \in \Delta^{T_c-1}$.

After substitution, we get two KL divergences as

$$\text{KL}(\widehat{\mathbf{C}}_{\mathbf{I}}^{\text{layout}} \| \mathbf{U}) = \sum_{i=1}^{T_c} \widehat{\mathbf{C}}_i^{\text{layout}} \log \left(\widehat{\mathbf{C}}_i^{\text{layout}} T_c\right),$$

*Table 3.* AUROC and AUPRC of the proposed method with varying RBO hyperparameters.

| Model | $p$ | SQuAD2.0 | | HotpotQA | | MS MARCO | |
|---|---|---|---|---|---|---|---|
| | | AUROC | AUPRC | AUROC | AUPRC | AUROC | AUPRC |
| LLaMA-3.2-3B-Instruct | 0.1 | 0.744 | 0.829 | 0.678 | 0.936 | 0.732 | 0.787 |
| | 0.3 | 0.742 | 0.828 | 0.673 | 0.935 | 0.729 | 0.782 |
| | 0.5 | 0.747 | 0.827 | 0.675 | 0.936 | 0.730 | 0.784 |
| | 0.7 | 0.735 | 0.855 | 0.672 | 0.934 | 0.727 | 0.778 |
| | 0.9 | 0.749 | 0.831 | 0.671 | 0.935 | 0.737 | 0.788 |
| LLaMA-3-8B-Instruct | 0.1 | 0.778 | 0.783 | 0.725 | 0.926 | 0.689 | 0.735 |
| | 0.3 | 0.778 | 0.782 | 0.727 | 0.927 | 0.689 | 0.735 |
| | 0.5 | 0.778 | 0.782 | 0.727 | 0.926 | 0.689 | 0.734 |
| | 0.7 | 0.778 | 0.782 | 0.727 | 0.927 | 0.690 | 0.737 |
| | 0.9 | 0.784 | 0.782 | 0.731 | 0.926 | 0.697 | 0.744 |
| Gemma-3-4B-it | 0.1 | 0.701 | 0.680 | 0.644 | 0.812 | 0.703 | 0.614 |
| | 0.3 | 0.702 | 0.679 | 0.641 | 0.812 | 0.705 | 0.623 |
| | 0.5 | 0.701 | 0.677 | 0.641 | 0.814 | 0.704 | 0.622 |
| | 0.7 | 0.703 | 0.684 | 0.645 | 0.814 | 0.706 | 0.627 |
| | 0.9 | 0.703 | 0.678 | 0.644 | 0.811 | 0.704 | 0.622 |

$$\mathrm{KL}(\widehat{\mathbf{P}}_{\mathbf{I}}^{\text{layout}} \,\|\, \mathbf{U}) = \sum\nolimits_{i=1}^{T_c} \widehat{\mathbf{P}}_i^{\text{layout}} \log\left(\widehat{\mathbf{P}}_i^{\text{layout}} T_c\right),$$

These formulations quantify the concentration of the layouts relative to a uniform distribution: higher KL values indicate that the layouts are more concentrated on a small subset of tokens, whereas lower KL values indicate a more uniform distribution of importance across tokens.

## D. Multi-level granularity

To capture a more comprehensive picture of how context information is processed within the model, we analyze emergence order and contribution layouts at multiple levels of granularity: token (subword)-level, word-level, and phrase-level.

**Token (subword)-level** provides the most fine-grained view, directly reflecting the internal representation of the model's vocabulary. Since many language models operate on subword units (e.g., Byte Pair Encoding or SentencePiece), examining this level allows us to trace how the model assembles meaning from its smallest representational units.

**Word-level** aggregates contributions and emergence orders across all subwords belonging to the same word. This reduces fragmentation introduced by subword tokenization, making the analysis more interpretable and directly comparable to human linguistic intuitions about words.

**Phrase-level** further groups words into coherent multi-word expressions. This clustering is conducted based on Shapley values in the relevance layout (Lundberg & Lee, 2017), which quantify each token's marginal contribution to the overall interpretation. By aggregating words that consistently share high relevance and interact strongly in terms of contribution, the phrase-level representation captures compositional semantics that cannot be observed at the word level alone. This granularity allows us to study how the model organizes meaning across larger linguistic units.

The three simulatability scores are measured at the token, word, and phrase levels, yielding nine features in total. Together with the two concentration scores and the overall context relevance score $r$, each sample is represented by a 12-dimensional feature vector for calibrator training.

## E. Inference Setup and Correctness Evaluation

To ensure the model focuses on short-form information-seeking questions, we instruct the model to generate responses containing at most five words. Specifically, we provide the model with an input sequence as follows:

> Answer the question in no more than five words.
> Context: {context} Question: {question} Answer:

Here, context and question are placeholders that are replaced with the retrieved passage and the corresponding query

from datasets, respectively. A concrete example of the input format is shown below.

> **Answer the question in no more than five words.**
> **Context:** Two Polish friends in Paris were also to play important roles in Chopin's life there. His fellow student at the Warsaw Conservatory, Julian Fontana, had originally tried unsuccessfully to establish himself in England; Albert Grzymala, who in Paris became a wealthy financier and society figure, often acted as Chopin's adviser and "gradually began to fill the role of elder brother in his life." Fontana was to become, in the words of Michalowski and Samson, Chopin's "general factotum and copyist."
> **Question:** What familial role was Albert Grzymała compared to in regard to Frédéric?
> **Answer:**

Since LLaMA-3.2-3B-Instruct, LLaMA-3-8B-Instruct, and Gemma-3-4B-it are relatively small models and may not reliably follow instructions, we further restrict decoding to a maximum of 10 generated tokens to encourage concise answers. Samples whose responses are not completed within this limit are discarded. In addition, to reduce variability caused by explanations, formatting artifacts, or trailing generations, we retain only the tokens appearing before the first period ("."), thereby extracting the model's earliest concise answer span for evaluation.

To determine if a predicted response is correct, we avoid token-level overlap metrics (e.g., BERTScore (Zhang et al., 2020)) and adopt a semantic evaluation pipeline. Specifically, we merge the predicted answer and the ground truth with the corresponding question into two natural language statements using Qwen2.5-7B (Yang et al., 2025). For example, given the input text below, suppose the model generates the statement "The capital of Washington state is Seattle."

> **Convert the following Q&A into a single factual sentence.**
> **Question:** Where is the capital of Washington state?
> **Answer:** Seattle.
> **Statement:**

We also construct a reference statement from the ground-truth answer "Olympia," namely, "The capital of Washington state is Olympia." The predicted statement is then evaluated using HHEM-2.1-Open (Bao et al., 2024). A prediction is labeled "incorrect" if its similarity score with the reference falls below 0.5. This approach assesses semantic correctness rather than surface token overlap, which is important because some questions admit multiple valid answers. For instance, for the question "When did World War II break out?", both "World War II broke out in 1939." and "World War II broke out in the late 1930s." are correct. This approach evaluates answer quality considering the question rather than relying on surface token overlap, and follows common practice in recent LLM evaluation pipelines (Qiu & Miikkulainen, 2024). Importantly, datasets contain unanswerable questions whose associated contexts lack the necessary information. In such cases, the QA model is expected to acknowledge the insufficiency of evidence explicitly. To evaluate this behavior, we compare the model's response against a set of predefined candidates (e.g., "I do not know.", "It is not mentioned.").

## F. Stability Analysis of UQ Methods

To ensure stable and reproducible results, we fixed the data sample split using a random seed (42) and performed all inference with greedy decoding. This setup eliminates variance from data partitioning and generation strategies, isolating the inherent stochasticity of the UQ methods themselves.

Among the baselines, only Semantic Entropy and Utility Ranker exhibit stochastic behavior. Semantic Entropy relies on an entailment model to cluster semantically similar generations, and the clustering introduces randomness. Utility Ranker retrieves distractor passages and uses contrastive learning to distinguish useful context, with the retrieval process being inherently stochastic. Our method also involves randomness due to the sampling-based estimation in the Shapley-value computation of token-level relevance layouts. We quantify the variance of our method, Semantic Entropy, and Utility Ranker across three independent runs for all model-dataset combinations, as reported in Table 4 and Table 5. From the results, we observe that all variances are extremely small (on the order of 1e-4 to 1e-8), indicating that the outcomes are highly stable.

## G. Discriminative ability

A natural concern is that the observed performance gains may primarily stem from the relevance score $r$, rather than from the proposed information-flow features (simulatability and concentration). To disentangle these effects, we conduct an ablation study comparing three settings: (i) our full information-flow features, (ii) a *relevance-score-only* setup, and (iii) a

*Table 4.* Stability analysis of UQ methods with inherent randomness. Reported values are AUROC means $\pm$ standard deviations.

| Model | Method | SQuAD2.0 | HotpotQA | MS MARCO |
|---|---|---|---|---|
| LLaMA-3.2-3B-Instruct | Semantic Entropy | 0.645 $\pm$ 2.87e-6 | 0.614 $\pm$ 5.83e-8 | 0.528 $\pm$ 8.04e-8 |
| | Utility Ranker | 0.629 $\pm$ 1.44e-6 | 0.597 $\pm$ 1.01e-5 | 0.593 $\pm$ 1.01e-4 |
| | Ours | 0.735 $\pm$ 7.23e-7 | 0.672 $\pm$ 8.61e-7 | 0.727 $\pm$ 1.01e-6 |
| LLaMA-3-8B-Instruct | Semantic Entropy | 0.515 $\pm$ 2.75e-6 | 0.493 $\pm$4.59e-6 | 0.518 $\pm$5.67e-4 |
| | Utility Ranker | 0.569 $\pm$ 6.20e-5 | 0.561 $\pm$ 9.48e-6 | 0.605 $\pm$6.33e-5 |
| | Ours | 0.778 $\pm$ 1.07e-4 | 0.727 $\pm$8.73e-5 | 0.690 $\pm$7.47e-6 |
| Gemma-3-4B-it | Semantic Entropy | 0.590 $\pm$ 2.10e-5 | 0.530 $\pm$ 8.67e-4 | 0.574 $\pm$ 1.41e-6 |
| | Utility Ranker | 0.643 $\pm$ 1.34e-4 | 0.545 $\pm$ 9.08e-5 | 0.564 $\pm$ 3.50e-5 |
| | Ours | 0.703 $\pm$ 3.87e-7 | 0.645 $\pm$ 3.42e-7 | 0.706 $\pm$ 5.22e-7 |

*Table 5.* Stability analysis of UQ methods with inherent randomness. Reported values are AUPRC means $\pm$ standard deviations.

| Model | Method | SQuAD2.0 | HotpotQA | MS MARCO |
|---|---|---|---|---|
| LLaMA-3.2-3B-Instruct | Semantic Entropy | 0.789 $\pm$ 2.99e-6 | 0.911 $\pm$ 1.98e-7 | 0.611 $\pm$ 7.19e-8 |
| | Utility Ranker | 0.789 $\pm$ 8.05e-6 | 0.905 $\pm$ 4.54e-7 | 0.661 $\pm$ 6.25e-5 |
| | Ours | 0.855 $\pm$ 3.65e-7 | 0.934 $\pm$ 5.75e-8 | 0.778 $\pm$ 1.75e-7 |
| LLaMA-3-8B-Instruct | Semantic Entropy | 0.542$\pm$3.22e-8 | 0.836$\pm$1.93e-4 | 0.582$\pm$8.90e-5 |
| | Utility Ranker | 0.588$\pm$7.54e-4 | 0.867$\pm$9.32e-5 | 0.650$\pm$6.98e-4 |
| | Ours | 0.782$\pm$5.37e-6 | 0.927$\pm$3.75e-4 | 0.737$\pm$6.01e-6 |
| Gemma-3-4B-it | Semantic Entropy | 0.546 $\pm$ 2.16e-5 | 0.727 $\pm$ 4.69e-6 | 0.479 $\pm$ 2.01e-4 |
| | Utility Ranker | 0.614 $\pm$ 1.55e-4 | 0.744 $\pm$ 2.27e-5 | 0.486 $\pm$ 4.80e-5 |
| | Ours | 0.684 $\pm$ 8.81e-8 | 0.814 $\pm$ 2.75e-8 | 0.627 $\pm$ 2.90e-6 |

*relevance-based concentration* variant, motivated by the intuition that when relevant tokens are more spatially concentrated in the context, the QA model is more likely to produce correct answers.

Table 6 summarizes the results, using Qwen-3-Reranker-8B as the ranker model. Across all datasets, the proposed information-flow features consistently outperform both (ii) and (iii), which rely solely on ranker-derived signals. This demonstrates that the performance improvements are not merely driven by the relevance score itself, but arise from modeling how information is propagated and utilized within the QA model.

While ranker-based features capture the overall relevance and, to some extent, the difficulty of the context, they remain agnostic to the model's internal computation. In contrast, our information-flow features explicitly characterize the model's reasoning dynamics, providing a more faithful account of how evidence is aggregated and leading to stronger predictive power in identifying model successes and failures.

*Table 6.* AUROC of the proposed information-flow features, ranker-score-only, and ranker-based-concentrated-only setups across datasets. Qwen-3-Reranker-8B is selected as the ranker model. Higher values are shown in **bold**.

| Model | Dataset | info-flow-features-only | relevance-score-only | ranker-based-concentration-only |
|---|---|---|---|---|
| LLaMA-3.2-3B-Instruct | SQuAD2.0 | **0.693** | 0.585 | 0.592 |
| | HotpotQA | **0.650** | 0.504 | 0.487 |
| | MS MARCO | **0.709** | 0.601 | 0.599 |
| LLaMA-3-8B-Instruct | SQuAD2.0 | **0.752** | 0.622 | 0.565 |
| | HotpotQA | **0.723** | 0.543 | 0.505 |
| | MS MARCO | **0.665** | 0.608 | 0.599 |
| Gemma-3-4B-it | SQuAD2.0 | **0.676** | 0.593 | 0.584 |
| | HotpotQA | **0.639** | 0.506 | 0.515 |
| | MS MARCO | **0.617** | 0.529 | 0.566 |

We further conduct an in-depth ablation to isolate the contributions of individual information-flow components, including the three Simulatability scores and the two Concentration scores. Specifically, for each metric, we perform a random

permutation at inference time to disrupt its signal while keeping all other components unchanged, and then measure the resulting degradation in AUROC and AUPRC.

The results in Figure 7 are averaged across all combinations of datasets and base QA models. Qwen-3-Reranker-8B is selected to estimate relevance layout. We observe that permuting any single metric consistently leads to a performance drop, indicating that each component contributes positively to the overall predictive performance. This confirms that the gains are not driven by a single feature, but rather arise from the complementary effects of multiple information-flow signals.

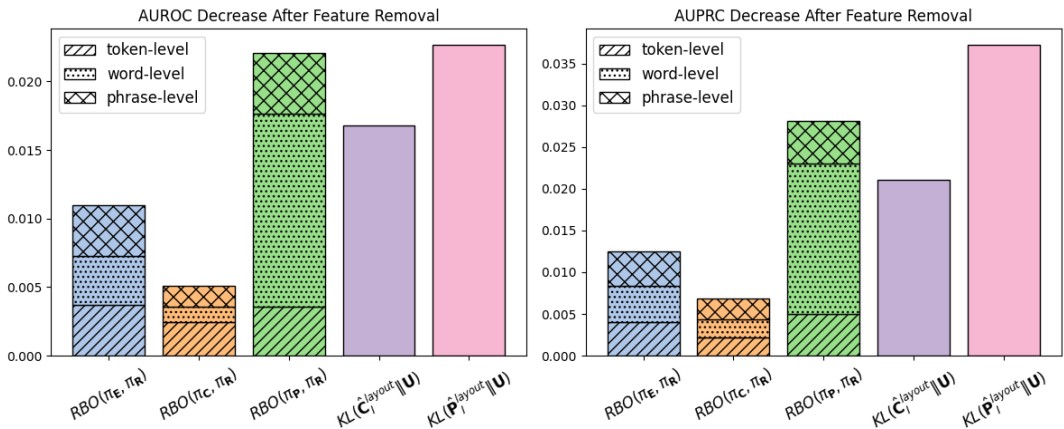

*Figure 7.* Impact of feature removal on model performance measured by AUROC (left) and AUPRC (right). Simulatability scores are organized by granularity level (token-, word-, and phrase-level). Concentration scores are based on KL divergence.

## H. Computational cost and applicability

**Computational cost**    A primary consideration for our method is the memory cost associated with computing the token contribution matrices $\mathbf{C}^{(l)} \in \mathbb{R}^{T \times T}$ for each layer $l$. The peak memory consumption occurs during the computation for a single layer. This step requires storing a raw, lower-triangular embedding matrix of size $T \times T \times d$, leading to a memory complexity of $O(T^2 d)$. Crucially, since intermediate embeddings for each layer can be discarded after processing, this peak memory cost is independent of the total number of layers $L$. We identify two strategies to mitigate this $O(T^2 d)$ cost:

(1) Low-Rank Approximation: The dimensionality $d$ of each vector in the matrix can be substantially reduced via projection, effectively lowering the $d$ factor in the $O(T^2 d)$ complexity.

(2) Sparse Storage: The token contribution matrices are typically lower-triangular and often exhibit sparsity, as many off-diagonal entries are negligible. Storing only the significant values can dramatically reduce the memory footprint.

After computation, storing the final projected matrices $\mathbf{C}^{(l)}$ for all $L$ layers requires only $O(T^2 L)$ memory. Given that $L \ll d$ in standard language model architectures, this cost is negligible compared to the peak computational overhead. This final storage requirement corresponds to the cost of computing the product of these matrices for the overall analysis.

**Applicability**    The major limitation of our method is its white-box nature, which requires access to internal model representations. Consequently, it is not directly applicable to closed-source large language models (LLMs) where such access is restricted. This reflects a fundamental trade-off between interpretability and universality.

Despite this, the value of our work is threefold. First, it provides a level of mechanistic insight into uncertainty that is unattainable with black-box and gray-box methods, offering a valuable benchmark for understanding the origins of model uncertainty. Second, it is immediately applicable to the growing suite of powerful open-source models, which are critical for academic research, safety auditing, and transparent deployments. Finally, the framework established here lays a foundation for future research into gray-box techniques that might approximate these information-theoretic measures with more limited access. Future work will focus on extending this paradigm. We aim to develop hybrid approaches that can operate effectively in gray-box settings and to explore distillation techniques to transfer the interpretability of white-box uncertainty estimates.

## I. Performance on calibration metrics

We evaluate methods that provide calibrated confidence, namely, P(True), KnowingMore, and our method, using calibration accuracy at a confidence threshold of 0.5 and the expected calibration error (ECE). As demonstrated in Table 7, our method also demonstrates the highest reliability among all evaluated approaches, achieving a calibration accuracy of 0.73. In addition, it attains the lowest expected calibration error (ECE) of 0.04, which is computed by partitioning the $[0, 1]$ confidence interval into 10 equal-width sub-intervals. These results indicate that the predicted confidence scores from our method are closely aligned with the empirical probability of correctness, reflecting not only accurate discrimination but also reliable confidence estimation.

*Table 7.* Uncertainty quantification performance of methods that provide calibrated confidence on three datasets, evaluated using calibration accuracy and expected calibration error (ECE). **Bold** denotes the best value for each metric, while the second-best is underlined.

| Model | Method | SQuAD2.0 Accuracy | SQuAD2.0 ECE | HotpotQA Accuracy | HotpotQA ECE | MS MARCO Accuracy | MS MARCO ECE | Mean Rank |
|---|---|---|---|---|---|---|---|---|
| LLaMA-3.2-3B-Instruct | P(True) | 0.533 | 0.161 | 0.666 | 0.303 | 0.579 | 0.103 | 2.83 |
| | KnowingMore | 0.653 | 0.222 | 0.877 | 0.106 | 0.611 | 0.136 | 2.17 |
| | Ours | **0.734** | **0.041** | **0.879** | **0.006** | **0.679** | **0.021** | 1.00 |
| LLaMA-3-8B-Instruct | P(True) | 0.623 | 0.159 | 0.706 | 0.199 | 0.569 | 0.260 | 2.67 |
| | KnowingMore | 0.600 | 0.247 | 0.836 | 0.133 | 0.584 | 0.142 | 2.33 |
| | Ours | **0.716** | **0.020** | **0.840** | **0.021** | **0.644** | **0.011** | 1.00 |
| Gemma-3-4B-it | P(True) | 0.508 | 0.438 | 0.319 | 0.575 | 0.568 | 0.359 | 3.00 |
| | KnowingMore | 0.590 | 0.259 | 0.707 | 0.181 | 0.582 | 0.131 | 2.00 |
| | Ours | **0.644** | **0.008** | **0.713** | **0.017** | **0.615** | **0.062** | 1.00 |

## J. Ranker model bias measurement

We verify the estimated relevance layouts from Qwen-3-Reranker-8B and measure the extent of bias from it. Specifically, for each dataset, we randomly sample 500 examples. Using the relevance layouts produced by Qwen-3-Reranker-8B as a reference, annotators inspect the top-ranked tokens for each example and label a layout "correct" if those top-ranked tokens were indeed helpful for answering the query. We retain only the examples with correct layouts. Then, we evaluate the simulatability metrics of LLaMA-3.2-3B-Instruct on both the retained subset and the original 500-sample collection for each dataset. The performance difference illustrate how much bias is introduced from the ranker model. AUROC and AUPRC of simulatability metrics on original and retained samples are shown in Table 8 and Table 9, which also list the percentage of samples retained after verification. We observe that AUROC and AUPRC increase slightly after human verification in general, indicating that the ranker's estimated relevance layouts contain some bias. This bias arises from the absence of true golden relevance layouts, rather than from our information-flow method. Moreover, the improvements are not significantly large, showing that the ranker remains a practical choice for automatically processing large-scale data.

*Table 8.* AUROC of simulatability metrics on the original 500 samples and the human-verified subset. **Bold** indicates higher values.

| | Metric | SQuAD2.0 Original | SQuAD2.0 Retained (94%) | HotpotQA Original | HotpotQA Retained (90%) | MS MARCO Original | MS MARCO Retained (84%) |
|---|---|---|---|---|---|---|---|
| token-level | $RBO(\pi_E, \pi_R)$ | 0.600 | **0.612** | 0.574 | **0.602** | 0.658 | **0.675** |
| | $RBO(\pi_C, \pi_R)$ | **0.603** | 0.590 | 0.534 | **0.535** | 0.657 | **0.668** |
| | $RBO(\pi_P, \pi_R)$ | **0.634** | 0.631 | 0.566 | **0.602** | 0.616 | **0.620** |
| word-level | $RBO(\pi_E, \pi_R)$ | 0.592 | **0.595** | 0.551 | **0.582** | 0.599 | **0.603** |
| | $RBO(\pi_C, \pi_R)$ | **0.551** | 0.550 | **0.530** | 0.518 | **0.613** | 0.611 |
| | $RBO(\pi_P, \pi_R)$ | 0.652 | **0.655** | 0.540 | **0.585** | 0.593 | **0.600** |
| phrase-level | $RBO(\pi_E, \pi_R)$ | 0.631 | **0.632** | 0.557 | **0.600** | 0.676 | **0.707** |
| | $RBO(\pi_C, \pi_R)$ | 0.621 | **0.630** | 0.575 | **0.580** | 0.706 | **0.726** |
| | $RBO(\pi_P, \pi_R)$ | 0.666 | **0.668** | 0.560 | **0.615** | 0.659 | **0.685** |

We further assess our reliance on the external ranker by replacing Qwen-3-Reranker-8B with the substantially smaller

*Table 9.* AUPRC of simulatability metrics on the original 500 samples and the human-verified subset. **Bold** indicates higher values.

| | Metric | SQuAD2.0 Original | SQuAD2.0 Retained (94%) | HotpotQA Original | HotpotQA Retained (90%) | MS MARCO Original | MS MARCO Retained (84%) |
|---|---|---|---|---|---|---|---|
| token-level | RBO($\pi_\mathbf{E}, \pi_\mathbf{R}$) | 0.931 | **0.942** | 0.914 | **0.929** | 0.700 | **0.781** |
| | RBO($\pi_\mathbf{C}, \pi_\mathbf{R}$) | 0.936 | **0.941** | 0.904 | **0.914** | 0.702 | **0.779** |
| | RBO($\pi_\mathbf{P}, \pi_\mathbf{R}$) | 0.939 | **0.945** | 0.903 | **0.929** | 0.675 | **0.744** |
| word-level | RBO($\pi_\mathbf{E}, \pi_\mathbf{R}$) | 0.934 | **0.944** | 0.900 | **0.918** | 0.645 | **0.715** |
| | RBO($\pi_\mathbf{C}, \pi_\mathbf{R}$) | **0.945** | 0.940 | 0.892 | **0.901** | 0.659 | **0.713** |
| | RBO($\pi_\mathbf{P}, \pi_\mathbf{R}$) | 0.949 | **0.955** | 0.893 | **0.923** | 0.643 | **0.718** |
| phrase-level | RBO($\pi_\mathbf{E}, \pi_\mathbf{R}$) | 0.942 | **0.948** | 0.908 | **0.929** | 0.720 | **0.794** |
| | RBO($\pi_\mathbf{C}, \pi_\mathbf{R}$) | 0.945 | **0.956** | 0.905 | **0.917** | 0.756 | **0.817** |
| | RBO($\pi_\mathbf{P}, \pi_\mathbf{R}$) | 0.952 | **0.958** | 0.902 | **0.930** | 0.707 | **0.777** |

MSMARCO-MiniLM-L12-v2 (Reimers & Gurevych, 2019) and BGE-v2-m3 (Chen et al., 2024b). As shown in Table 10, our method maintains competitive performance without substantial degradation, indicating limited sensitivity to the choice of ranker.

*Table 10.* Comparison of the proposed UQ method's performance when using different ranker models.

| Model | Ranker | SQuAD2.0 AUROC | SQuAD2.0 AUPRC | HotpotQA AUROC | HotpotQA AUPRC | MS MARCO AUROC | MS MARCO AUPRC |
|---|---|---|---|---|---|---|---|
| LLaMA-3.2 3B-Instruct | MiniLM-L12-v2 (33.4M) | 0.711 | 0.839 | 0.634 | 0.923 | 0.695 | 0.751 |
| | BGE-v2-m3 (0.6B) | 0.752 | 0.864 | 0.649 | 0.926 | 0.723 | 0.771 |
| | Qwen-3-Reranker-8B | 0.735 | 0.855 | 0.672 | 0.934 | 0.727 | 0.778 |
| LLaMA-3 8B-Instruct | MiniLM-L12-v2 (33.4M) | 0.757 | 0.769 | 0.712 | 0.918 | 0.675 | 0.717 |
| | BGE-v2-m3 (0.6B) | 0.780 | 0.792 | 0.717 | 0.921 | 0.689 | 0.734 |
| | Qwen-3-Reranker-8B | 0.778 | 0.782 | 0.727 | 0.927 | 0.690 | 0.737 |
| Gemma-3 4B-it | MiniLM-L12-v2 (33.4M) | 0.664 | 0.648 | 0.651 | 0.817 | 0.683 | 0.602 |
| | BGE-v2-m3 (0.6B) | 0.718 | 0.701 | 0.639 | 0.817 | 0.696 | 0.626 |
| | Qwen-3-Reranker-8B | 0.703 | 0.684 | 0.645 | 0.814 | 0.706 | 0.627 |

# K. Effect of post-hoc calibration

The inclusion of a calibrator in our method could lead to the question of whether performance gains stem from the proposed metrics or the post-processing. To address this, we designed our evaluation to dissociate these two factors. We equipped baselines whose original formulations do not involve training with the same calibration procedure used in our method.

We report results for both their raw scores and their calibrated variants in Table 11, Table 12, and Table 13. The results show that the calibrated variants of these baselines do not outperform their raw versions, and in most cases perform even worse due to overfitting. This outcome is expected: these baselines inherently produce a single scalar uncertainty score (e.g., Perplexity, Semantic Entropy) that is intrinsically designed to correlate with prediction error monotonically. In other words, their discriminative power is largely "built-in". Applying a calibrator to such one-dimensional signals offers no new information and, as our results show, often degrades performance through overfitting.

In contrast, our method and other multi-dimensional approaches (e.g., Utility Ranker, KnowingMore) generate a spectrum of complementary indicators. Conventional evaluation frameworks like AUROC, which require a single scalar, are inherently ill-suited to assess these multi-dimensional signals directly. The post-hoc model is therefore not a performance-enhancing "calibrator" but a necessary scalarization function. Its role is to project the rich, multi-faceted information from our metrics onto a single axis for fair comparison. Thus, this step is not a privileged addition but a fundamental requirement to make multi-dimensional UQ methods evaluable against their scalar counterparts.

*Table 11.* Comparison of raw and calibrated baseline variants using LLaMA-3.2-3B-Instruct as the QA model. Higher values are in **bold**.

| | | SQuAD2.0 | | HotpotQA | | MS MARCO | |
| --- | --- | --- | --- | --- | --- | --- | --- |
| | | AUROC | AUPRC | AUROC | AUPRC | AUROC | AUPRC |
| PPL | Raw | **0.608** | **0.795** | 0.582 | **0.912** | **0.592** | **0.691** |
| | Cali. | 0.607 | 0.785 | **0.583** | 0.910 | 0.590 | 0.684 |
| P(True) | Raw | **0.546** | **0.746** | 0.467 | 0.872 | **0.557** | **0.664** |
| | Cali. | 0.545 | 0.735 | **0.488** | **0.877** | 0.500 | 0.592 |
| Regular Entropy | Raw | **0.698** | **0.829** | **0.651** | **0.924** | **0.654** | **0.729** |
| | Cali. | 0.697 | 0.827 | 0.649 | 0.920 | 0.651 | 0.722 |
| Semantic Entropy | Raw | 0.645 | 0.789 | 0.614 | **0.911** | **0.528** | **0.611** |
| | Cali. | **0.646** | **0.791** | **0.617** | 0.910 | 0.521 | 0.606 |
| Attention Score | Raw | **0.513** | **0.718** | 0.478 | 0.867 | 0.459 | 0.561 |
| | Cali. | 0.506 | 0.712 | **0.506** | **0.882** | **0.534** | **0.618** |
| Focus | Raw | **0.703** | **0.830** | **0.701** | **0.944** | **0.690** | **0.752** |
| | Cali. | 0.702 | 0.828 | 0.699 | 0.940 | 0.663 | 0.739 |

*Table 12.* Comparison of raw and calibrated baseline variants using Gemma-3-4B-it as the QA model. Higher values are in **bold**.

| | | SQuAD2.0 | | HotpotQA | | MS MARCO | |
| --- | --- | --- | --- | --- | --- | --- | --- |
| | | AUROC | AUPRC | AUROC | AUPRC | AUROC | AUPRC |
| PPL | Raw | **0.639** | **0.622** | **0.605** | **0.772** | **0.561** | **0.495** |
| | Cali. | 0.638 | 0.617 | 0.604 | 0.768 | 0.551 | 0.487 |
| P(True) | Raw | **0.545** | **0.521** | **0.525** | **0.725** | **0.548** | **0.484** |
| | Cali. | 0.500 | 0.493 | 0.500 | 0.712 | 0.500 | 0.479 |
| Regular Entropy | Raw | **0.658** | **0.633** | 0.617 | **0.779** | 0.570 | 0.505 |
| | Cali. | 0.657 | 0.629 | **0.618** | 0.778 | **0.587** | **0.511** |
| Semantic Entropy | Raw | **0.590** | **0.546** | 0.530 | 0.727 | **0.574** | **0.479** |
| | Cali. | 0.500 | 0.494 | **0.533** | **0.728** | 0.473 | 0.413 |
| Attention Score | Raw | **0.529** | **0.518** | 0.477 | 0.698 | 0.526 | 0.451 |
| | Cali. | 0.512 | 0.502 | **0.520** | **0.733** | 0.526 | **0.453** |
| Focus | Raw | 0.653 | 0.636 | **0.650** | **0.832** | **0.574** | **0.519** |
| | Cali. | **0.654** | **0.643** | 0.644 | 0.827 | 0.573 | 0.516 |

*Table 13.* Comparison of raw and calibrated baseline variants using LLaMA-3-8B-Instruct as the QA model. Higher values are in **bold**.

| | | SQuAD2.0 | | HotpotQA | | MS MARCO | |
| --- | --- | --- | --- | --- | --- | --- | --- |
| | | AUROC | AUPRC | AUROC | AUPRC | AUROC | AUPRC |
| PPL | Raw | 0.600 | 0.636 | 0.531 | 0.857 | 0.529 | 0.605 |
| | Cali. | **0.619** | **0.640** | **0.568** | **0.869** | **0.530** | 0.605 |
| P(True) | Raw | 0.676 | 0.687 | 0.622 | **0.886** | **0.554** | **0.626** |
| | Cali. | **0.704** | **0.697** | **0.643** | 0.885 | 0.550 | 0.609 |
| Regular Entropy | Raw | 0.649 | **0.656** | **0.583** | **0.873** | 0.568 | **0.624** |
| | Cali. | **0.650** | 0.653 | 0.581 | 0.869 | 0.568 | 0.620 |
| Semantic Entropy | Raw | **0.515** | 0.542 | 0.493 | 0.836 | **0.518** | **0.582** |
| | Cali. | 0.512 | 0.542 | **0.501** | **0.840** | 0.502 | 0.572 |
| Attention Score | Raw | 0.493 | 0.535 | 0.492 | 0.831 | 0.485 | 0.559 |
| | Cali. | **0.520** | **0.548** | **0.519** | **0.848** | **0.509** | **0.576** |
| Focus | Raw | 0.719 | **0.724** | 0.703 | **0.916** | 0.693 | **0.726** |
| | Cali. | 0.719 | 0.720 | 0.703 | 0.913 | **0.694** | 0.724 |

