# OpenReview forum: "Information Flow Reveals When to Trust Language Models"
_ICML.cc/2026/Conference — ICML 2026 spotlight_

### Official Review · Reviewer_DtvD · 2026-02-22

**Soundness:** 3
**Presentation:** 3
**Significance:** 2
**Originality:** 3
**Overall Recommendation:** 4
**Confidence:** 4

**Summary:**

The authors propose a method to decompose the effects of each input token of a language model on the model's predicted output and apply it for the task of uncertainty quantification in a RAG setting. They posit that model correctness correlates with a) a sharp distribution of token influence (a high "concentration") and b) high correlation between predicted token contributions and ground-truth token relevance scores. They evaluate the method on several RAG benchmarks and find that resulting confidence scores outperform baselines from the UQ literature.

**Compliance With Llm Reviewing Policy:**

Affirmed.

**Final Justification:**

The rebuttal addressed my concerns and I raised my score.

**Key Questions For Authors:**

How exactly are individual tokens ablated in the faithfulness experiment?

**Limitations:**

The authors include the required impact statement but do not explicitly discuss the limitations of the method.

**Strengths And Weaknesses:**

Strengths:

The paper is fairly clearly written and easy to follow. The authors include several helpful ablation studies, including an experiment suggesting that the tokens identified as most influential by the method do in fact affect the model's predictions. I'd say the method is fairly inventive and differs conceptually from most prominent UQ baselines, most importantly by not requiring an expensive sampling step. The reported benchmark results seem quite strong.

Weaknesses:

Though the paper has an extensive set of experiments, one of the most important ones is absent. The method produces two scores, only one of which is wholly derived from the "information flow" technique that's novel to this paper. The other ("simulatability") is derived from "ground truth" labels generated by a reranking model. Using automated techniques to generate ground-truth labels for experiments usually only makes sense when said automated techniques are impracticably slow or otherwise limited; this is clearly not the case here. I'd like to see how well the ranker alone would perform on the same benchmarks to get a better idea of what the information flow is contributing (in theory, it should even be possible to derive a "concentration" score based entirely on the ranker, without using information flow. That'd be a useful addition too). By the same token, it would be interesting to see which of the values thrown into the bag (pi_C, pi_E, pi_R) are most important.

One of the main downsides of methods like semantic entropy is the expensive sampling step, which renders the method completely unusable in most practical applications. This method appears to have the advantage of not requiring that, but details about the method's runtime are not provided in the paper, and so this is difficult to adjudicate.

---

> ### Author Rebuttal · Authors · 2026-03-29
>
> We thank the reviewer for their insightful comments and address the concerns as follows.
> >The method produces two scores, only one of which is wholly derived from the "information flow" technique that's novel to this paper. The other ("simulatability") is derived from "ground truth" labels generated by a reranking model.
>
> We want to clarify that simulatability is not solely derived from the ranker. Instead, it is defined as a consistency measure between the information-flow-based signals and the ranker output. Therefore, simulatability is fundamentally a **joint signal**, rather than a standalone ranker-based feature.
>
> >Using automated techniques to generate ground-truth labels for experiments usually only makes sense when said automated techniques are impracticably slow or otherwise limited; this is clearly not the case here.
>
> Our use of the ranker is not to replace readily available ground-truth relevance layout, but to approximate fine-grained token-level relevance, for which **human annotation would be prohibitively expensive**. In particular, assessing the usefulness of each context token with respect to a query requires detailed human judgment and does not scale to large datasets. To address this, we use a ranker combined with SHAP to provide a **scalable and consistent proxy** for token-level relevance.
>
> >I'd like to see how well the ranker alone would perform on the same benchmarks to get a better idea of what the information flow is contributing (in theory, it should even be possible to derive a "concentration" score based entirely on the ranker, without using information flow. That'd be a useful addition too).
>
> In **Table 6**, we compared the performance of the proposed information-flow features with the relevance score $r$. Following the reviewer’s suggestion, we further developed a ranker-based concentration score, based on the intuition that when relevant tokens are more concentrated in the context, the QA model is more likely to respond correctly. The **Table** ([link](https://anonymous.4open.science/r/Anonymized-info-flow-EDA0/table_ranker_concentration.png)) extends Table 6 by incorporating this ranker-based concentration metric.
>
> While the ranker-based features can reflect the overall relevance and difficulty of the context, they do not capture how the QA model actually processes this information. In contrast, our information-flow features provide insights into the model’s internal reasoning and are more effective at explaining model errors.
>
> > By the same token, it would be interesting to see which of the values thrown into the bag (pi_C, pi_E, pi_R) are most important.
>
> We performed ablation studies by randomly permuting each metric individually during inference, then measuring the resulting drop in AUROC and AUPRC. The results in the **Figure**([link](https://anonymous.4open.science/r/Anonymized-info-flow-EDA0/fig_feature_importance.png)) are averaged over all combinatory settings of datasets, QA models, and ranker models, showing that **each proposed metric contributes positively to the overall performance**. We also note that concentration scores contribute relatively more, likely because bias in the rankers limits the full effectiveness of simulatability scores (see Appendix I).
>
> >How exactly are individual tokens ablated in the faithfulness experiment?
>
> After identifying the top- and bottom-ranked tokens based on each proposed measure, we remove these tokens from the input. This directly tests their influence on the model’s prediction by eliminating their information entirely.
>
> >The authors do not explicitly discuss the limitations of the method.
>
> We investigated the potential bias from ranker models in **Appendix I**, and discussed memory cost and applicability in **Appendix G**.
>
> >One of the main downsides of methods like semantic entropy is the expensive sampling step, which renders the method completely unusable. This method appears to have the advantage of not requiring that, but details about the method's runtime are not provided in the paper, and this is difficult to adjudicate.
>
> Computing the Manhattan-distance-based contribution for a single token pair in Eq. (4) requires roughly $3d$ FLOPs, accounting for the absolute difference, sum, and max operations. For a sequence of $T$ tokens, considering all lower-triangular pairs in a single layer, this amounts to approximately $1.5 T^2 d$ FLOPs per layer. Across Llayers, the total cost scales as $1.5 L T^2 d$. The subsequent multiplication of the $L$ dense contribution matrices requires roughly $2(L-1)T^3$FLOPs.
>
> For instance, with 1,000 tokens on Llama-3.2-3B-Instruct (3,072-dimensional embeddings and 24 layers), computing the Manhattan distances requires on the order of $1.1×10^{11}$ FLOPs, while multiplying the 24 dense $1,000×1,000$ matrices takes roughly $4.6×10^{10}$ FLOPs. On a 10 TFLOPS GPU, this results in a latency of approximately **16 ms per generated token**, which is negligible for most NLP applications.

---

> > ### Author Rebuttal · Reviewer_DtvD · 2026-04-03
> >
> > Thanks for the high-effort rebuttal. I've raised my score to a 4.

---

> > > ### Author Response · Authors · 2026-04-03
> > >
> > > We sincerely appreciate the reviewer's feedback. We are delighted that our rebuttal has addressed the concerns and that the reviewer raised the score to a positive one. The suggestion to ablate each proposed feature is highly insightful, and we will add the corresponding results to the paper.

---

### Official Review · Reviewer_Vr2z · 2026-02-24

**Soundness:** 3
**Presentation:** 2
**Significance:** 3
**Originality:** 3
**Overall Recommendation:** 4
**Confidence:** 3

**Summary:**

Uncertainty quantification in RAG remains challenging, as existing methods often overlook how LLMs internally process and integrate contextual information. This paper proposes a novel framework that models the layer-wise information flow within LLMs to assess the contribution of each input token to the generated output. Two key metrics are introduced—Emergence Order and Contribution Layout—which capture the propagation and concentration of contextual information across layers. These are further combined with external relevance signals to train a calibrator that estimates response confidence. Experiments on multiple benchmarks demonstrate that the proposed approach outperforms existing methods in both discriminative power and cross-dataset generalization.

**Compliance With Llm Reviewing Policy:**

Affirmed.

**Final Justification:**

The authors' response has further reinforced my previous judgment, so I would like to maintain the positive rating of "weak accept."

**Key Questions For Authors:**

1. Can the design rationale behind the two quantities—Emergence Order and Contribution Layout—be explained in an intuitive and accessible manner?

2. Compared to traditional "result-oriented" approaches (e.g., logits, entropy), what essential new insights does this "process-oriented" perspective offer for understanding when and why RAG systems make mistakes?

**Limitations:**

yes

**Strengths And Weaknesses:**

**Strengths:**
1. The paper introduces a conceptually novel angle by leveraging internal information flow for UQ, moving beyond surface-level features like logits or attention.
2. Experiments across multiple models and datasets, with ablation studies and stability analysis, support the claims.
3. The ablation study provides strong evidence that the identified information flows are functionally relevant.

**Weaknesses:**
1. The memory complexity is non-trivial and may limit applicability to long sequences or resource-constrained settings.
2. The method requires access to internal representations, making it inapplicable to closed-source LLMs. This is acknowledged but not addressed
3. The motivation behind the proposed method is not immediately clear. For instance, the choice of introducing quantities such as Emergence Order and Contribution Layout lacks an intuitive explanation in the introduction. While these concepts can be understood through a careful reading of the paper, the lack of accessibility undermines the overall readability.

---

> ### Author Rebuttal · Authors · 2026-03-28
>
> We thank the reviewer for their insightful comments and address the concerns as follows.
>
> **Weakness 1**
> >The memory complexity is non-trivial...
>
> In **Appendix G**, we provide a detailed analysis of memory usage. Our method’s memory scales roughly as $O(T^2 d)$, where $T$ is the input length and $d$ is the model’s hidden dimension. Since the contribution matrix at each layer is lower-triangular, only $(T(T+1))/2$ elements need to be stored per hidden dimension. For example, with $T=2000$ and $d=2048$, this corresponds to approximately $2000⋅2001/2⋅2048≈4.1×10^9$ elements. Using bf16 precision (2 bytes per element), this results in roughly 8.2 GB of memory, which is well within the capacity of modern hardware and practical for typical NLP applications such as document-level reasoning or summarization.
>
> **Weakness 2**
> >The method requires access to internal representations...
>
> We acknowledge that our method requires access to internal representations, which makes it inapplicable to closed-source LLMs. Nonetheless, studying white-box models is crucial, as it allows us to directly probe how the model processes information and provides insights into its decision-making. This approach represents a frontier in uncertainty quantification research for LLMs and is an essential step toward understanding and improving model reliability.
>
> **Weakness 3 and Question 1**
> >The motivation behind the proposed method is not immediately clear...Can the design rationale behind the two quantities...
>
> Emergence Order and Contribution Layout capture two complementary perspectives on how information flows from input tokens to a model’s prediction.
>
> **Emergence Order** is computed by tracing backward from the final-layer decisive representation of the last input token.  Tokens that appear earlier in this backward traversal are those whose information is incorporated more directly and prominently into the final decision, and can therefore be interpreted as **primary evidence for the model’s output**.
>
> **Contribution Layout**, in contrast, aggregates influence over all admissible paths through the network, including indirect and multi-hop interactions mediated by intermediate tokens and layers. This allows it to capture tokens that may not serve as direct evidence for the prediction but are nonetheless essential in **building a comprehensive understanding of the context**. Their influence accumulates across the network, making the layout more distributed.
>
> For example, consider the sentence:
>
> *“Alice gave Bob the book after she finished reading it,”*
>
> where the task is to resolve *“she,”* and the model correctly predicts *“Alice.”* Emergence Order highlights direct, decision-critical tokens such as *“she”* and *“Alice”* in the context, whereas Contribution Layout also assigns importance to tokens like *“book”* and *“reading,”* which help establish the semantic context.
>
> To provide an intuitive explanation, we can use a **traffic analogy**. Imagine all tokens as cities and the model as a road network carrying traffic toward a final destination (the prediction). Emergence Order asks which cities’ traffic appears first when traced backward from the destination. These cities are the primary sources driving the final outcome. Contribution Layout considers all roads in the network, measuring the total traffic from each city that eventually reaches the destination, even if it passes through multiple intermediate cities. This illustrates how Emergence Order captures **direct, decision-critical influence**, while Contribution Layout also considers **distributed, context-building influence**.
>
> We thank the reviewer for the comment and will add an intuitive explanation in the revised manuscript.
>
> **Question 2**
> >Compared to traditional "result-oriented" approaches...
>
> Compared to traditional “result-oriented” approaches such as logits or entropy, which only measure the confidence a model holds about its prediction, the process-oriented perspective provides insight into **how the model arrives at that prediction**. Our method traces the flow of information from individual tokens to the final decision, identifies which tokens act as primary evidence or indirect support, and detects when reasoning pathways are misleading. This allows us to understand **why RAG systems make mistakes**, not just when they do, and can highlight structural vulnerabilities, such as reliance on spurious tokens, that result-oriented metrics cannot reveal.

---

> > ### Author Rebuttal · Reviewer_Vr2z · 2026-04-01
> >
> > I'd like to thank the authors for responding my questions in detail. I have no more questions and I'd like to maintain my positive rating.

---

> > > ### Author Response · Authors · 2026-04-02
> > >
> > > We sincerely thank the reviewer for their positive feedback and for taking the time to carefully consider our revisions. These valuable inputs has helped us strengthen the paper. We are delighted to hear that our responses addressed all concerns.

---

### Official Review · Reviewer_ed1N · 2026-03-04

**Soundness:** 3
**Presentation:** 3
**Significance:** 3
**Originality:** 3
**Overall Recommendation:** 4
**Confidence:** 3

**Summary:**

This paper uses information flow to build a layer-wise trace that reveals each context token’s contribution to the output, providing an interpretable basis for assessing reliability. From this analysis, it introduces two measures to calibrate prediction confidence. The first, simulatability, posits that a prediction is more likely to be correct when context token contributions align closely with their true relevance. The second, concentration, asserts that a response is more likely to be correct when it is derived from a narrow, focused subset of tokens.

**Compliance With Llm Reviewing Policy:**

Affirmed.

**Final Justification:**

I'd like to keep my positive rating about this paper.

**Key Questions For Authors:**

1. Does your method work well on LLMs with large sizes (such as 7B and 13B)?
2. What is the importance of Emergence Order and Contribution Layout, respectively?
3. Why does your method perform worse than baselines on HotpotQA? Is there any explanation?

**Limitations:**

yes

**Strengths And Weaknesses:**

### Strengths
1. Detailed demonstration of the method.
2. Motivated design of the metrics.
3. Improved performance.

### Weaknesses
1. The scale of the LLMs is limited.
2. More experiments can be adopted to improve the reliability of the metrics.

---

> ### Author Rebuttal · Authors · 2026-03-30
>
> We thank the reviewer for their insightful comments and address the concerns as follows.
> **Weakness 1 and Question 1**
> > The scale of the LLMs is limited. Does your method work well on LLMs with large sizes (such as 7B and 13B)?
>
> We evaluate our method on a larger-scale model, LLaMA-3-8B-Instruct, using the SQuAD 2.0 dataset. The results are shown in **Table**([link](https://anonymous.4open.science/r/Anonymized-info-flow-EDA0/table_llama-3-8B.png)), where our method consistently outperforms all baselines in both AUROC and AUPRC. This demonstrates that the proposed approach scales effectively to larger models.
>
> **Weakness 2**
> > More experiments can be adopted to improve the reliability of the metrics.
>
> To assess the reliability of our metrics, we conducted a stability analysis in **Appendix E**, demonstrating that the experimental results are robust and reproducible across different runs.
>
> To further evaluate the actual usefulness of each proposed metric, we performed ablation studies by randomly permuting each metric individually during inference and measuring the resulting drop in AUROC and AUPRC. The results in the **Figure** ([link](https://anonymous.4open.science/r/Anonymized-info-flow-EDA0/fig_feature_importance.png)), averaged over all combinations of datasets, QA models, and ranker models, show that each metric contributes positively to the overall performance.
>
> **Weakness 3**
> >What is the importance of Emergence Order and Contribution Layout, respectively?
>
> Emergence Order and Contribution Layout capture two complementary perspectives on how information flows from input tokens to a model’s prediction.
>
> **Emergence Order** is computed by tracing backward from the final-layer decisive representation of the last input token.  Context tokens that appear earlier in this backward traversal are those whose information is incorporated more directly and prominently into the final decision, and can therefore be interpreted as **primary evidence for the model’s output**.
>
> **Contribution Layout**, in contrast, aggregates influence over all admissible paths through the network, including indirect and multi-hop interactions mediated by intermediate tokens and layers. This allows it to capture context tokens that may not serve as direct evidence for the prediction but are nonetheless essential in **building a comprehensive understanding of the context**. Their influence accumulates across the network, making the layout more distributed.
>
> For example, consider the sentence:
>
> *“Alice gave Bob the book after she finished reading it,”*
>
> where the task is to resolve *“she,”* and the model correctly predicts *“Alice.”* Emergence Order highlights direct, decision-critical tokens such as *“she”* and *“Alice”* in the context, whereas Contribution Layout also assigns importance to tokens like *“book”* and *“reading,”* which help establish the semantic context.
>
> >Why does your method perform worse than baselines on HotpotQA? Is there any explanation?
>
> Emergence Order and Contribution Layout are particularly well-suited to settings where predictions are supported by the retrieval of relevant evidence, such as factual or context-grounded tasks. However, HotpotQA contains **multi-hop reasoning scenarios**, where correct predictions often rely on intermediate logical operations, including entity comparison, and implicit inference steps that are not localized to specific tokens. These reasoning processes may not be embedded by token-level contributions.
>
> As a result, the proposed metrics are not always optimal for datasets emphasizing complex multi-step reasoning. Nevertheless, despite this task mismatch, **our method still outperforms most baselines on HotpotQA**, suggesting that capturing information flow and grounding remains broadly effective even in settings that require deeper reasoning.

---

> > ### Author Rebuttal · Reviewer_ed1N · 2026-04-03
> >
> > Thanks for the rebuttal, I'd like to keep my positive rating about this paper.

---

> > > ### Author Response · Authors · 2026-04-03
> > >
> > > We are glad to know that our responses addressed all concerns. We thank the reviewer for the insightful comments and constructive feedback.

---

### Official Review · Reviewer_Z214 · 2026-03-08

**Soundness:** 3
**Presentation:** 3
**Significance:** 3
**Originality:** 3
**Overall Recommendation:** 5
**Confidence:** 3

**Summary:**

The paper proposes a novel uncertainty quantification framework that leverage information flow to quantify the importance of context tokens to the generated response.
The authors first defined Emergence order and Contribution layout to measure the relative importance of input tokens to the output based on information flow, and further defined two prediction confidence features, Simulatability and Concentration.
Specifically, Simulatability is computed through comparing the Emergence order with the Relevance Layout, which is inferred by a LLM ranker and Shapley framework.
The method finally trains an XGBoost calibrator on the proposed new features to output calibrated confidence for the model’s response.
The experiments evaluated the proposed method against extensive baseline methods on two small-scale LLMs and three benchmarks, and revealed that the proposed method produces more reliable uncertainty estimates.
The further ablation studies demonstrate that the proposed method is robust to distribution shifts, the identified top-rank tokens are crucial to the predictions, the bias of ranker has limited impact on the performance, and the performance improvements are due to the proposed metrics rather than the post-hoc calibration.

**Compliance With Llm Reviewing Policy:**

Affirmed.

**Final Justification:**

The rebuttal addressed my concerns. I raised my score to 5.

**Key Questions For Authors:**

**Key Questions For Authors:**
1. How is the Shapley framework implemented? Does it function independently from the LLM ranker that assign a scalar score r representing the overall relevance of a context to a given question?
2. In algorithm 1: whenever assign a new emergence order to index j, does it need to remove all $C\_{i',j}^{k'}$ with the same j in the S? Otherwise, it is likely that in a later iteration argmax(S) will return another $C\_{i',j}^{k'}$ with the same j and assign a new Emergence order to j, which is unwanted.
3. In line 374 right column “we randomly selected 100 correctly predicted samples from each experimental configuration.” It is unclear about “correctly predicted samples”. Are they sampled from the cases where LLMs generate correct results according to HHEM-2.1-Open? Or are they sampled from the cases where the proposed uncertainty metric correctly predicted the correctness?

**Limitations:**

The authors didn’t discuss the limitation of  their work in the paper. It might be worth to reflect on the design pitfalls of Emergence Order, Contribution Layout, simulatability and concentration

**Strengths And Weaknesses:**

**Strengths:**
1. The paper designs novel prediction confidence features that leverage information flow through the models, which are insightful
2. The evaluation covers extensive baselines and demonstrated that the proposed method has significant improvement in uncertainty estimation, underscoring the value of incorporating the entire information propagation process
3. The experiments in Sec.6 are systematic, and the findings about out-of-distribution generalizability, causal faithfulness, ranker bias and post-hoc calibration further validate the effectiveness of the proposed method
4. The paper is well structured with clear descriptions of the proposed metrices and informative figures

**Weaknesses:**
1. The proposed Emergence order relies on the principal information flows, which are extracted in an iterative process (algorithm 1). At each iteration, the algorithm selects the maximum $C^{(l)}\_{i,j}$  in $S$. However, the $C^{(l)}\_{i,j}$ represents the contribution matrix of layer $l$ for $(y\_i, x\_j)$, which is normalized at layer $l$ (according to equation 4). $S$ would contain $C^{(l)}\_{i,j}$ of different layers. It seems unfair to compare $C^{(l)}\_{i,j}$ from different layers and pick the absolute maximum $C^{(l)}\_{i,j}$ from S as the principal information flows without additional normalization across different layers. The actual importance of the identified principle flows is unclear.
2. The authors claim that self-contributions are always dominant due to residual connections so that, after identifying the maximum $C^{(l)}\_{i,j}$ in each iteration, the algorithm directly treats the flow from $x\_{j}$ at layer $l$ down straight to the $x\_{j}$ at layer $1$ as the principal information flow. The claim is strong but does not have sufficient explanation.
3. Sec 4.4.3 writes “we combine these features—simulatability, concentration, and the context relevance score r—to train a calibrator”. But the paper introduced three Simulatability scores ($RBO(\pi\_C, \pi\_R)$, $RBO(\pi\_P, \pi\_R)$, and $RBO(\pi\_E, \pi\_R)$) and two Concentration scores (KL(C|U), KL(P|U)). It is unclear about how and which of these five scores were actually used in the experiments.
4. The paper does not conduct in-depth separation analysis on the effect of the three Simulatability scores and the two Concentration scores. It is hard to infer which of these five scores has the most impact on the performance of uncertainty prediction. And it is unclear whether each metric actually helps or affect negatively the uncertainty quantification.

Minor:
- Error in equation (6): Is the second \sum intended to be s2=j…s3?

---

> ### Author Rebuttal · Authors · 2026-03-28
>
> We thank the reviewer for their insightful comments and address the concerns as follows.
>
> **Weakness 1**
> >The proposed Emergence order relies on the principal information flows...
>
> After the layer-wise normalization in Equation (4), each row of $\mathbf{C}^{(l)}$ sums to 1, i.e.,
> $\sum_{j=1}^T \mathbf{C}_{i,j}^{(l)} =1$ for all $i$ and $l$.
> In other words, for **each target token** $y_i$ at **any layer** $l$, the contributions from all source tokens $x_1,…,x_T$ form a normalized distribution. This ensures that contribution values are on a consistent scale across layers, making them directly comparable when selecting principal information flows. This normalization scheme follows prior work [1], which similarly compares normalized contribution matrices across layers.
>
> [1] Ferrando, J., G´allego, G. I., and Costa-Juss`a, M. R. Measuring the mixing of contextual information in the transformer. arXiv preprint arXiv:2203.04212, 2022.
>
> **Weakness 2**
> >The authors claim that self-contributions...
>
> The dominance of self-contributions can be understood from the design of transformer architectures. Residual connections parameterize each layer as an additive update rather than a full transformation, encouraging the model to learn small, incremental refinements around an identity mapping. As a result, representations are iteratively refined across layers rather than overwritten [2]. Pre-normalization further constrains the scale of per-layer updates while allowing the residual stream to accumulate changes across depth, which naturally leads to strong self-contributions [3].
>
> This design intuition is consistent with our empirical observations: even when residual connections are included in the selection pool  $S$ instead of being directly incorporated into the principal information flow, they are consistently selected in the earliest iterations. We thank the reviewer for the comment and will add a clarification in the revised manuscript.
>
> [2] Jastrzebski, Stanislaw et al. “Residual Connections Encourage Iterative Inference.” ArXiv abs/1710.04773 (2017).
>
> [3] Xiong, Ruibin et al. “On Layer Normalization in the Transformer Architecture.” ArXiv abs/2002.04745 (2020).
>
> **Weakness 3**
>
> >Sec 4.4.3 writes “we combine these features...
>
> The three simulatability scores are measured at the token, word, and phrase levels, yielding nine features in total, as mentioned in **Appendix C**. Together with the two concentration scores and the overall context relevance score $r$, each sample is represented by a 12-dimensional feature vector for calibrator training.
>
> **Weakness 4**
>
> >The paper does not conduct in-depth separation...
>
> We conduct ablation studies by randomly permuting each metric at inference time, thereby disrupting its signal, and then measure the resulting drop in AUROC and AUPRC. The results in the **Figure**([link](https://anonymous.4open.science/r/Anonymized-info-flow-EDA0/fig_feature_importance.png)) are averaged over all combinatory settings of datasets, QA models, and ranker models, showing that each proposed metric contributes positively to the overall performance.
>
> **Question 1**
>
> >How is the Shapley framework...
>
> The overall relevance of a context to a given question is computed independently by the ranker model, which assigns a scalar score $r$. We then apply the SHAP framework [4] on top of this ranker to quantify how individual tokens contribute to the overall relevance score. In other words, SHAP is used to measure the effect of removing or perturbing tokens on the ranker’s output, without altering the underlying ranker.
>
> [4] Lundberg, Scott M. and Su-In Lee. “A Unified Approach to Interpreting Model Predictions.” NIPS 2017.
>
> **Question 2**
> >In algorithm 1: whenever assign...
>
> The concern is valid. In the practical implementation of Algorithm 1, we ensure that once a context token is assigned an order, it cannot receive a new order in subsequent iterations. We will also update the algorithm description to explicitly include this step.
>
> **Question 3**
> >In line 374 right column “we randomly ...
>
> The “correctly predicted samples” refer to instances where the LLM generated the correct answers according to HHEM-2.1-Open. We sample these cases so that, when some tokens are removed, we can directly evaluate whether the QA model is still able to produce the correct prediction, investigating the effect of the token removal.
>
> **Question 4**
> >The authors didn’t discuss the limitation...
>
> One potential limitation of our work is the bias introduced by the ranker model, which directly affects the reliability of the simulatability metric. **Appendix I** shows that ranker-based annotation, while not as precise as human labels, is fairly reliable for our experiments. We also discuss the memory cost and the applicability of our method in **Appendix G**.
>
> **Typo**
> >Error in equation (6): Is the second \sum intended to be s2=j…s3?
>
> We thank the reviewer for pointing out the typo. We will correct it in our work.

---

> > ### Author Rebuttal · Reviewer_Z214 · 2026-04-02
> >
> > Thanks for the clear and detailed response. I have no more questions.
> >
> > In my opinion, your response to Weakness 4 is an important piece of evidence showing the effectiveness of the designed 11 features. Can you confirm that given acceptance, you will add this new ablation study to the main text as a (sub)section or appendix (but clearly mentioned it in the main text)? Please also add clarification for Weakness 2&3 and Question 3. I will raise my score to 5.

---

> > > ### Author Response · Authors · 2026-04-02
> > >
> > > We sincerely thank the reviewer for their constructive feedback. We confirm that, upon acceptance, we will add the response corresponding to Weakness 2, 3, 4 and Question 3 to the script as the reviewer suggested. These additions will improve the clarity and completeness of our presentation.

---

### Decision · Program_Chairs · 2026-04-30

**Decision:**

Accept (spotlight)

**Comment:**

The paper is viewed very positively overall by the reviewers and myself. The core idea, using internal information flow to quantify uncertainty in RAG, is novel, technically interesting, and well motivated. The proposed notions of Emergence Order and Contribution Layout, the resulting simulatability and concentration features, and the strong empirical performance against competitive baselines are particularly strong aspects of this paper. The experimental section was generally seen as thorough, with ablations and analyses that support the method’s faithfulness, robustness, and usefulness beyond standard output-based confidence measures.

The main concerns are relatively limited in scope: the initial presentation left some design choices insufficiently intuitive, the evaluation was at first somewhat constrained in model scale, and reviewers asked for clearer separation of the contribution of the different features, as well as more detail on runtime, memory, and implementation. The rebuttal appears to have addressed these points effectively, including additional experiments and clarifications, and this led reviewers to maintain or strengthen their positive assessments. Overall, this is a technically solid and distinctive contribution that is likely to be of interest to the community.